

# Brief Communications: Tides and Damage as Drivers of Lake Drainages on Shackleton Ice Shelf

Julius Sommer[1], Maaike Izeboud[2], Sophie de Roda Husman[2], Bert Wouters[2], and Stef Lhermitte[2,3]

[1]Department of Chemical Engineering, Delft University of Technology, Delft, 2628CN, The Netherlands
[2]Department of Geoscience & Remote Sensing, Delft University of Technology, Delft, 2628CN, The Netherlands
[3]Department of Earth & Environmental Sciences, KU Leuven, Leuven, B-3001, Belgium

**Correspondence:** Maaike Izeboud (m.izeboud@tudelft.nl)

**Abstract.** To investigate the drivers of lake drainages in Antarctica, we analyzed optical remote sensing data from the Shackleton Ice Shelf in East Antarctica over three melt seasons. Our study identified one drainage event in 2018-2019, eleven in 2019-2020, and one in 2020-2021. All identified drainages occurred in regions with medium to high levels of ice shelf damage and with active damage development. Additionally, 12 out of 13 drainages coincided with increases in tidal heights. These findings provide insights into the factors influencing current lake drainages in Antarctica and may help in predicting future drainage events.

## 1 Introduction

Surface lake drainage can destabilize ice shelves, occurring either slowly via supraglacial channels or rapidly through crevasses driven by the weight of the water – a process known as hydrofracture (Nye and Perutz, 1957). While Greenland's lake drainages are relatively well-studied (Williamson et al., 2018; McMillan et al., 2007), much less is known about similar processes in Antarctica. Understanding when and how surface lakes drain is crucial for assessing ice shelf stability. Trusel et al. (2022) have shown that lake drainages on the Amery Ice Shelf are linked to high-amplitude tidal cycles. However, it remains unclear which lakes drain across Antarctica and whether this relationship with tidal range is a continent-wide phenomenon.

In Antarctica, where most surface lakes form near the grounding line, only some of these lakes drain. We hypothesize that this variability is related to the structural integrity of the ice shelf. Specifically, we propose that both the amount of damage (open crevasses, fractures and rifts) and the "activeness" of the damage development on the ice shelf – referring to the orientation of damage relative to ice flow as indication of their opening mode – may influence lake drainage events. High activeness could lead to increased fracturing due to tensile stresses, whereas in less active regions, fractures may close due to compressive stresses. Specifically, lakes on undamaged, non-active ice shelves may simply refreeze, while those on moderately damaged, active shelves are more likely to drain due to preconditioned fractures that allow meltwater to flow and further destabilize the ice (Lai et al., 2020). On severely damaged ice shelves, where extensive fracturing has already compromised the structure, lakes may not form at all.

The Shackleton Ice Shelf presents an ideal case for further investigation, as it features both draining and non-draining lakes, alongside regions of varying structural damage (Arthur et al., 2020; de Roda Husman et al., 2023). The key questions here



are: Where do observations of meltwater ponding and damage overlap? Which meltwater ponds are draining, and under what conditions? Can draining events be traced back to specific triggers, such as the destabilization of pre-existing fractures or external forces like tidal flexing? Addressing these questions on the Shackleton Ice Shelf will help clarify the mechanisms behind surface lake drainage and their role in ice shelf destabilization, contributing to a broader understanding of Antarctic ice shelf dynamics.

In this study, we first detect surface lakes on the Shackleton Ice Shelf for the melt seasons of 2018-2019, 2019-2020, and 2020-2021 using a threshold-based method developed by Moussavi et al. (2020) based on optical imagery. We then identify drainage events by analyzing whether meltwater lakes have disappeared between consecutive images. These drainage events are compared to damage and activeness data of the Shackleton Ice Shelf produced by Izeboud and Lhermitte (2023). Additionally, we examine the timing of the drainage events in relation to tidal height using a tidal model (Padman et al., 2002). By integrating

these data, we aim to reveal how ice shelf conditions influence the dynamics of surface lake drainage.

## 2   Materials and Methods

This study utilizes multi-source satellite imagery and a tidal height simulation model to investigate rapid lake drainages on ice shelves. We employ detection methods for meltwater, lake drainages, and damage, integrating their outputs to analyze spatial distribution patterns. Satellite image access and initial processing are conducted through the Google Earth Engine (GEE)

platform.

### 2.1   Satellite Imagery

Optical imagery is used to identify supraglacial lakes and drainage events. Sentinel-2 Level-1C (S2) and Landsat 8 Collection 2 Tier 2 TOA reflectance (L8) images are assessed via GEE and filtered by a maximum cloud coverage of 30 % and minimum sun elevation of 20 °, following a similar approach as (Tuckett et al., 2021). Median image mosaics are produced over time

periods of 8 days for L8 and 10 days for S2, with resolutions of 30 m and 10 m, respectively. For the 2018-2019, 2019-2020, and 2020-2021 melt seasons, each November to March, the mosaics are sorted by their assigned date which is defined as halfway between first and latest date of all containing images. For optimized subsequent processing, the optical imagery was processed using a grid, focusing on an area that covers 95 % of the Shackleton Ice Shelf, as defined by Natural Earth (2017).

  Synthetic aperture radar (SAR) data is utilized for damage detection and characterization. S1 images (instrument mode:

"EW" and polarization: "HH") are used, with one median mosaic per year collected between 1 and 10 November to minimize misinterpretation of contrast differences due to meltwater.

  Antarctic ice flow velocity observations of 2019 are obtained from the ITS_LIVE campaign (Gardner et al., 2020). The grounding line and ice shelf front are obtained from Gerrish et al. (2022) and kept constant during the studied period.



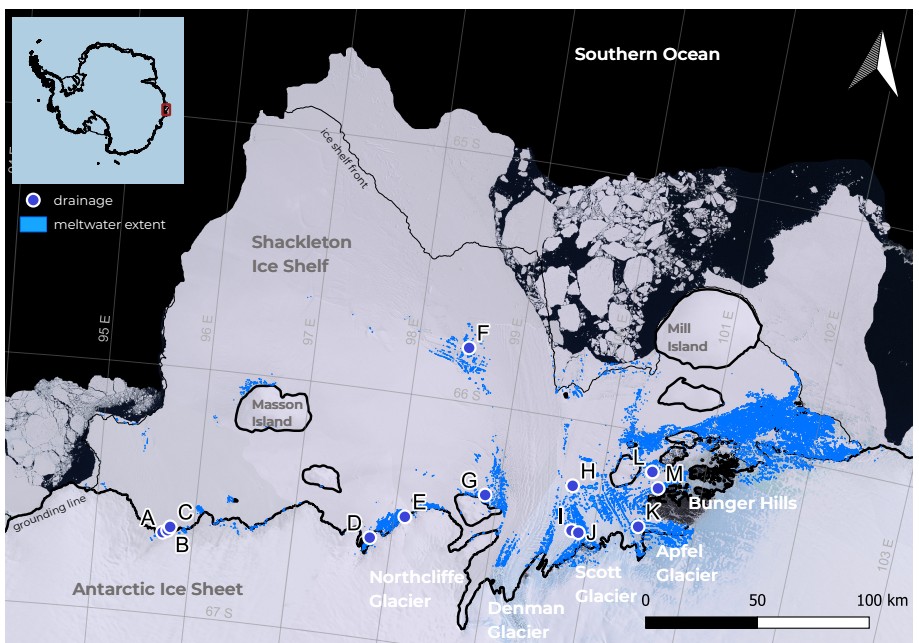

**Figure 1.** Geographic overview of the Shackleton Ice Shelf study area. Background shows LIMA by Bindschadler et al. (2008). Inset shows location in East Antarctica. Blue: Maximum meltwater extent 2019/20 based on S2 and L8 observations. Thick black line: Grounding line. Thin black line: Shelf coastline. Both from MEaSURES data set by Gerrish et al. (2022).

## 2.2 Supraglacial Lake Detection and Lake Volume Calculation

The location and depth of supraglacial lakes are determined using a threshold-based method. Moussavi et al. (2020) developed the method and thresholds, which were further refined and automated by Tuckett et al. (2021). Their method involves combining different bands of the satellite images, applying thresholds to obtain masks for meltwater, rocks and clouds, and finally estimating the depth by taking into account albedo and reflectance of water. These steps are executed within GEE, and produce meltwater lake masks and meltwater depth output. The lake volume is then obtained by multiplying the depths with

their respective pixel areas (Pope et al., 2016). Outliers are removed in a subsequent step if they are not located on ice mass, have a misinterpreted depth of less than $0\,\mathrm{m}$, or have a surface area of less than $1800\,\mathrm{m}^2$ as suggested by Williamson et al. (2018). A maximum lake extent for each melt season is obtained by combining all lake masks of a season.

## 2.3 Lake Drainage Detection

To identify lake drainages, the optical time series (Section 2.1) are analyzed. A drainage event is identified if a supraglacial

lake (Section 2.2) lost at least $80\,\%$ of its area between two consecutive images, provided that no more than 10 days elapsed between those images (Arthur et al., 2020).



Further, we only retain those lakes that meet the following conditions: surface area greater than $54000\,\mathrm{m}^2$ (Williamson et al., 2017), median lake depth greater than $0.65\,\mathrm{m}$, standard deviation of lake depth greater than 0.3, and distance to the nearest masked cloud greater than $500\,\mathrm{m}$. These filtering steps focus on large, deep lakes with a defined depth profile, while avoiding misinterpretations due to faulty cloud masks or blue ice regions.

As a final step, we perform a manual visual inspection of the detected drainage events utilizing all available non-mosaic imagery from L8, and S2. Twelve out of twenty-five events are removed, as they are judged to be refreezing lakes rather than draining lakes.

### 2.4 Damage Detection and Activeness

The Normalised Radon transform Damage detection (NeRD) method (Izeboud and Lhermitte, 2023) is used to detect and analyze damage features on the ice shelf from strong linear contrasts in Sentinel-1 SAR backscatter images (line detection). The S1 images are resampled to the same L8 resolution of $30\,\mathrm{m}$ and processed with a window size of 10 pixels within the NeRD algorithm, yielding damage maps of $300\,\mathrm{m}$ with a damage signal between 0 and 0.5 (low to high damage indication) and their orientation (-90 to $90°$).

The obtained damage orientation is used to identify areas with a likelihood of active damage development, by comparing the damage orientation to local ice flow angle. First, the ice flow velocity was resampled to match the resolution of the damage orientation map. Then, an area was considered likely to be 'active' if the damage orientation and flow angle have a difference of $45°$ (tolerating a deviation of $15°$), which occurs mainly in the shear zones of the ice shelf or for mixed mode opening fractures. We exclude large open rifts near the ice front that are perpendicular to the ice flow, cut completely through the ice and hence irrelevant for lake drainages, as well as other orientations likely representing older, inactive fractures misaligned with current stress fields (Colgan et al., 2016). This metric represents a binary mask that marks those pixels whose damage orientation relative to the flow direction of the ice suggests damage under active development.

Both $300\,\mathrm{m}$ rasters of damage (0 to 0.5) and activeness (binary, 0 or 1) are then downsampled with a factor of 10 using an average resampling method and normalized with their respective maxima, resulting in $3000\,\mathrm{m}$ rasters with values between 0 and 1. This process allows for an indication of how much a local lake drainage event is influenced by the characteristics of its surrounding area. Finally, the rasters are clipped to the ice shelf, masking out any values on the sea and on the grounded ice.

### 2.5 Tidal Heights

To investigate how ocean tides influence drainage events, we utilize the Circum-Antarctic Tidal Simulation (CATS2008) model to compute daily tidal amplitudes. These amplitudes are determined by calculating the difference between the maximum and minimum tidal heights for each day. The CATS2008 model is a high-resolution barotropic tide model specifically designed for the Antarctic continental shelf, incorporating bathymetry data from various sources and assimilating tide gauge and satellite altimetry observations to enhance its accuracy. The model solves the depth-integrated shallow water equations on a finite difference grid, accounting for the effects of sea ice cover and under-ice shelf cavities, which are crucial for accurately representing tidal dynamics in polar regions (Howard et al., 2019; Padman et al., 2002).





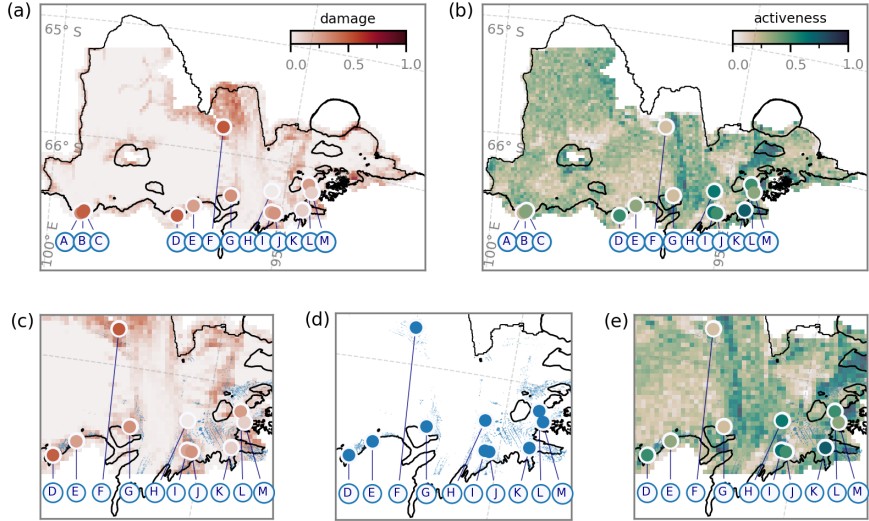

**Figure 2.** Spatial occurrence of lake drainages on Shackleton ice shelf. a) and b) show NeRD damage and activeness of summer 2019/20 as derived in subsection 2.4. The bottom row shows a zoomed section around the glaciers on the ice shelf, with c) NeRD damage d) maximum lake extent, and e) activeness of summer 2019/20. Coloured dots with blue labels: Drainage events found over the Antarctic summer of 2018/19, 2019/20 and 2020/21. Colour of the dot represents the underlying pixel quality for the respective summer. Thick black line: Grounding line. Thin black line: Shelf coastline. Both from from MEaSURES data set by Gerrish et al. (2022).

## 3 Results

During the study period, extensive supraglacial lake extents were detected, with maxima of $257\,\mathrm{km}^2$ for 2018-2019, $261\,\mathrm{km}^2$ for 2019-2020, and $197\,\mathrm{km}^2$ for 2020-2021. Notably, the second period, summer 2019-2020 was historically recognized as a record summer (Robinson et al., 2020). The meltwater primarily accumulated near the grounding line and between Bunger Hills and Mill Island, a pattern consistent with previous studies such as those by de Roda Husman et al. (2023); Arthur et al. (2020). Saunderson et al. (2022) also noted significant melt in the north of the West-Shackleton ice shelf. However, the findings indicate minimal lake formations in this region, suggesting immediate drainage of meltwater into the ocean in highly damaged areas (Figure 1 and 2).

Based on these lake masks (subsection 2.2) and lake drainage detection method (subsection 2.3), we identify thirteen lake drainage events during the study period: one in the 2018-2019 melt season, eleven in the 2019-2020 melt season, and one in the 2020-2021 melt season (Figure 2). The eleven events during the 2019-2020 melt season further exemplify the extreme conditions of that Antarctic summer. Refer to Table A1 for a detailed list of all detected events.





## 3.1 Lake Drainage Events in Areas of the Ice Shelf with Medium-to-High Damage or Activeness

The NeRD damage value highlights the spatial distribution of damage features on the ice, with high values indicating areas characterized by rifts or crevasses. Building on the findings of Lai et al. (2020), this study demonstrates how the extent and severity of damage influence the distribution of lake drainages. On the Shackleton Ice Shelf, damage levels can be categorized into three distinct groups: not/low damaged (values < 0.01, 64 % of the area, no lake draining events), medium-damaged (0.01 < values < 0.20, 28 % of the area, three lake draining events), and highly damaged (values > 0.20, 8 % of the area, ten lake draining events). For a detailed listing of the classification and damage distribution, refer to Table B1 and Figure B1.

The results show that lake drainages predominantly occur in areas classified as medium to highly damaged (Figure 2a and 2c). The three lake drainages in medium-damage regions are particularly concentrated around Scott Glacier and Bunger Hills (drainages H, L, and M). Although high-damage areas cover only a small portion of the Shackleton Ice Shelf, they account for the majority of lake drainages, primarily located further west along the grounding line (drainages A to E), with one event occurring on the glacier tongue (drainage F). In contrast, no lake drainages have been recorded in low-damage areas, despite these areas covering the majority of the ice shelf.

In addition to the damage value, the NeRD algorithm also provides information on the dominant orientation of detected fractures, from which the activeness of the ice shelf is derived (Figure 2b and 2e). The activeness parameter provides insights into the dynamic behavior of the ice, with high values indicating areas where the ice is actively deforming. Given that activeness values are distributed in a bell-shaped curve across the ice shelf (refer to Figure B1), other thresholds than for the damage metric were used to categorize the ice into distinct groups: not active (values < 0.12, 10 % of the area), moderately active (0.12 < values < 0.40, 71 % of the areas), and highly active (0.40 < values, 19 % of the area), as detailed in Table B1.

Regions with high activeness are observed around the glaciers, as well as around the northern tip of West Shackleton, and along the grounding lines of both West and East Shackleton, refer to Figure 1 and 2b. In contrast to the damage metric, this activeness tends to concentrate inland, away from the ice shelf edge. The inward concentration of high activeness is due to how activeness is calculated, reflecting the orientation of damage relative to ice flow. When damage aligns at approximately 45 ° to the direction of ice flow, this configuration becomes more prominent. This pattern occurs where the moving ice experiences shear stress from adjacent slower-moving or stationary ice masses, as described by Colgan et al. (2016). Glacier zones, where ice flow is most pronounced, serve as the clearest examples of areas exhibiting high activeness.

Unlike damage values, the activeness parameter does not appear to be directly related to the distribution of accumulated meltwater (Figure 2d and 2e). However, all of the detected lake drainage events occur in areas of the ice shelf classified as medium to highly active (seven and six lake draining events respectively, Figure 2b and Table B1). This observation suggests that while activeness may not directly influence meltwater distribution, it plays a role in the occurrence of drainage events. The distribution of the events indicates that lake drainages are more likely to occur in areas with moderate to high activeness, highlighting the role of ice dynamics in the behavior of the Shackleton Ice Shelf.

Considering both damage and activeness, several notable patterns emerge. First, the grounding line on West Shackleton (drainages A to E) serves as a prototypical example where both damage and activeness are high at the lake drainage sites. This



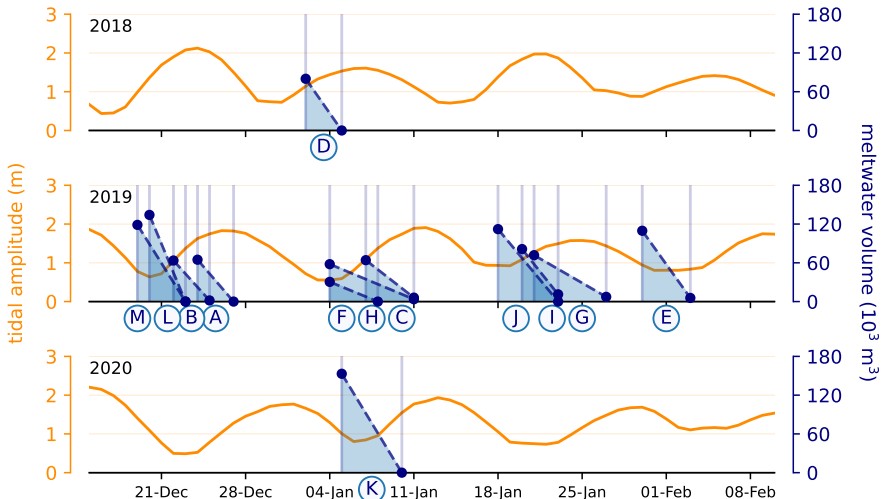

**Figure 3.** Time-windows of individual drainages and CATS2008 tidal amplitudes. Blue dots and shaded area: Change of meltwater volume for individual lakes (subsection 2.2).

indicates that areas along the grounding line are prone to such events due to the combined effect of significant damage and high activeness. Second, east of the glacier tongue (drainages H to M), lake drainage events occurred in areas of high activeness, despite relatively low levels of damage. This suggests that the interaction between the small glaciers and the Bunger Hills creates a highly active section of the ice shelf, where accumulated meltwater rapidly drains even in moderately fractured regions.

Intriguingly, activeness and damage don't always exhibit parallel trends, suggesting a more complex influence of these two factors on lake drainages. For example, drainages H, K, and M took place in areas with relatively low damage but high activeness, while drainages A, F, and G occurred in the least active regions, yet showed high levels of damage. By integrating information on the dominant orientation of fractures relative to ice flow direction, we identify with the activeness metric an independent characteristic of ice shelf damage.

Overall, it can be concluded that both damage and activeness are important indicators of the likelihood of accumulated meltwater draining rapidly. Although the exact relationship between these metrics requires further investigation across different ice shelves and with more drainage events, it is clear that at least one of these factors, damage or activeness, must be present for lake drainage to occur.

**3.2    Lake Drainage Events in Periods of Increasing Tidal Heights**

Our findings unveil a compelling narrative of ice shelf dynamics, revealing an intricate interplay between tidal forces and supraglacial lake drainage events. Building upon the concept introduced by Trusel et al. (2022) on the Amery Ice Shelf, we present evidence supporting a tidal mechanism that determine the timing of lake drainages on the Shackleton Ice Shelf.





Comparison of drainage time-windows and tidal data indicates a pattern: drainage episodes consistently align with the ascending phase of tide cycles, with one exception across all observed events (Figure 3, with drainage E considered as exception).

This synchronicity between tidal movements and lake drainage initiation suggests more than mere coincidence; it points to a causal relationship that reshapes our understanding of ice shelf processes. Given the tidal amplitudes and the geometry of the Shackleton Ice Shelf, large areas likely experience cyclical stress states driven by tidal oscillations (Padman et al., 2002). As tidal amplitudes surge towards their spring maxima, the ice shelf undergoes enhanced flexure, potentially generating complex tensile stress fields at both its surface and base.

Our data reveals a trend: Lake drainage events occur more frequently as tidal amplitudes increase. This pattern suggests that amplified tidal flexure may serve as a catalyst, either initiating new cracks or reactivating dormant weaknesses within the ice shelf structure. Once this process begins, the rapid influx of draining water could further propagate these fractures, facilitating complete lake drainage in a cascade of events reminiscent of those described by Das et al. (2008).

These observations, encompassing multiple drainage events, lend strong support to a model where tidal forcing plays a pivotal role in modulating supraglacial lake drainage across the ice shelf. This mechanism, particularly potent in regions subject to significant tidal flexure, not only confirms but also extends the work of Trusel et al. (2022).

## 4    Discussion and Conclusion

The study of meltwater drainage on the Shackleton Ice Shelf reveals a complex interplay of factors influencing ice shelf hydrology. Our findings highlight the multifaceted nature of drainage events and provide a framework for understanding their spatial and temporal distribution.

Drainage of accumulated meltwater results from an intricate combination of ice shelf damage, damage activeness, meltwater accumulation, and tidal forces. This complexity underscores the need for a holistic approach when studying ice shelf hydrology. While our study provides possible new insights, it is important to recognize that the proposed explanation is still a simplified model of a highly complex system.

One of the primary difficulties in studying ice shelf hydrology is detecting drainage events. Rapid drainages are often indicative of hydrofracture, which occurs within a few hours to days when the accumulation of meltwater into existing fractures or the formation of new cracks leads to sudden drainage. The rapid nature of these drainages means they can occur and conclude between satellite observations, making it challenging to capture the precise moment of hydrofracture initiation. While it is possible to detect the aftermath of drainage events, assigning these events specifically to hydrofracture becomes difficult. Furthermore, rapid filling and drainage events that occur between image acquisitions may go undetected, cloud cover and other atmospheric disturbances can obscure satellite imagery and potentially mask drainage events, and the temporal resolution of satellite passes may not be sufficient to capture all drainage events, especially those of short duration. The small number of observed events in our study poses challenges for statistical analysis and may not fully represent the true frequency of drainage occurrences. This limitation underscores the need for higher temporal resolution in satellite data or the integration of complementary observational methods to accurately identify and attribute rapid drainage events to hydrofracture mechanisms.



In terms of methodology, the NeRD method does not detect individual fractures but instead identifies areas of damage, which is useful in indicating zones of weakness across the ice shelf. This approach aligns well with our focus on larger-scale weakening associated with drainage events. However, it does limit a more physical representation of fractures, suggesting that for future research, a more detailed analysis of individual fractures could provide deeper insights.

Similarly, the activeness parameter offers a simplified approach by comparing fracture orientations to the local velocity field. This method is suitable for our study and complements the damage detection approach by identifying areas of weakening that coincide with lake drainage. Nevertheless, with the insights gained here, further investigation into how fracture development and stability align with observed lake drainages would be valuable. Methods such as linear elastic fracture mechanics (e.g., Lai et al. (2020)) could enhance our understanding of how fractures evolve and influence drainage processes.

While normalizing damage and activeness values based on their maxima aids comparison, it can introduce biases. These biases may overemphasize or underplay certain regions, depending on the distribution of values across the ice shelf. Furthermore, the thresholds used to categorize damage and activeness were specifically tailored for this study, which could produce different results if applied to other regions. The NeRD algorithm's reliance on thresholds also emphasizes the importance of carefully selecting and adjusting these criteria when applying it to different ice shelves.

Downsampling the damage and activeness rasters by a factor of 10 (to $3000\,\text{m}$) enabled us to identify meaningful relationships that were not apparent at the original $300\,\text{m}$ resolution produced by the NeRD algorithm. The initial high-resolution analysis failed to show clear patterns, likely because it captured too much localized variation, obscuring broader trends. Resampling to a coarser scale revealed the relationships between damage, activeness, and drainage events. However, the low activeness observed in drainages F and G, despite their proximity to glacial zones where higher activeness was expected due to shear forces, raises questions about the chosen downsampling factor or assumptions about activeness in these regions. This activeness metric offers a promising new perspective on drainage dynamics but requires further refinement and validation to improve its predictive power.

Given the complex nature of drainage events and the challenges in their detection, there is a clear need for more sophisticated statistical approaches. Future research should focus on developing probabilistic models that can account for the uncertainties in drainage detection and the influence of various factors, and are able to identify subtle patterns in ice shelf characteristics that may precede drainage events, and conducting time series analyses to better understand the temporal dynamics of meltwater accumulation and drainage.

When expanding our study to an Antarctic-wide scale, we could assess if the link between drainage events and medium damage/high activeness/large tidal range occurs on larger scale. The combination of damage severity, activeness, and meltwater accumulation could serve as indicators for potential drainage sites. However, the development of a reliable predictive model would require validation of the activeness metric across multiple ice shelves and melt seasons, integration of real-time or near-real-time data on meltwater accumulation and ice shelf damage, and consideration of additional factors such as ice shelf thickness, basal melting rates, and local climate conditions.



*Code and data availability.* Data and scripts will be made available on GitHub and Zenodo following the initial review process.



## Appendix A:  Detected Lake Drainage Events

**Table A1.** Detected lake drainages on the Shackleton Ice Shelf during the Antarctic summers of 2018-2019, 2019-2020, and 2020-2021. Refer to subsection 2.3 and 2.4 for detailed detection methodologies.

| event | lon | lat | draining period | | lake volume change in m$^3$ | | damage in - | activeness in - |
|---|---|---|---|---|---|---|---|---|
| A | 95.6880194462098 | -66.6712014092243 | 24.12.2019 | 27.12.2019 | 64800 | - | 0.331 | 0.203 |
| B | 95.7372639879535 | -66.6609833484217 | 22.12.2019 | 25.12.2019 | 63582 | 1743 | 0.477 | 0.362 |
| C | 95.7593493354464 | -66.6468296358281 | 07.01.2020 | 11.01.2020 | 64086 | 6092 | 0.477 | 0.362 |
| D | 97.7817520694010 | -66.5949656409632 | 02.01.2019 | 05.01.2019 | 80024 | - | 0.478 | 0.513 |
| E | 98.1040021313363 | -66.4923153429390 | 30.01.2020 | 03.02.2020 | 109621 | 5646 | 0.265 | 0.338 |
| F | 98.4935779957272 | -65.7826184475536 | 04.01.2020 | 08.01.2020 | 30502 | - | 0.504 | 0.173 |
| G | 98.8703252460483 | -66.3577772279784 | 21.01.2020 | 27.01.2020 | 71799 | 7308 | 0.315 | 0.185 |
| H | 99.7193031471739 | -66.2642446799923 | 04.01.2020 | 11.01.2020 | 57902 | 3637 | 0.015 | 0.616 |
| I | 99.7845357662211 | -66.4428832719498 | 20.01.2020 | 23.01.2020 | 81276 | 11201 | 0.238 | 0.656 |
| J | 99.8553409670035 | -66.4465678857868 | 18.01.2020 | 23.01.2020 | 112135 | - | 0.307 | 0.498 |
| K | 100.4363173489130 | -66.3803925791929 | 05.01.2021 | 10.01.2021 | 153085 | - | 0.105 | 0.727 |
| L | 100.4768799867120 | -66.1531862562149 | 20.12.2019 | 23.12.2019 | 134149 | - | 0.274 | 0.505 |
| M | 100.5615613327310 | -66.2140012167503 | 19.12.2019 | 23.12.2019 | 118717 | - | 0.102 | 0.344 |

## Appendix B:  Damage and Activeness

**Table B1.** Spatial distribution of lake drainage events classified by damage and activeness. Classification thresholds are given by the lower limit. The table shows the fraction of the Shackleton Ice Shelf affected and the number of lake drainage events in each category. Refer to subsection 2.4 for details on damage and activeness classifications.

| | damage | | | activeness | | |
|---|---|---|---|---|---|---|
| | threshold | area | events | threshold | area | events |
| high | 0.20 | 0.08 | 10 | 0.40 | 0.19 | 6 |
| medium | 0.01 | 0.28 | 3 | 0.12 | 0.71 | 7 |
| low | - | 0.64 | - | - | 0.10 | - |



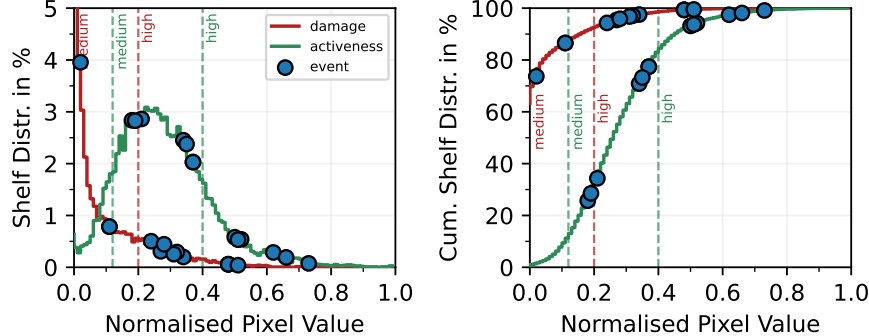

**Figure B1.** Activeness and damage distribution on the Shackleton Ice Shelf, including the distribution of lake drainages listed in Table A1. The limits are indicated as introduced in Table B1. Refer to subsection 2.4 for details on damage and activeness classifications.

*Author contributions.* J.S.: Conceptualization, Methodology, Visualization, Writing – original draft, Data curation. M.I. and S.R.H.: Conceptualization, Methodology, Writing – review & editing. B.W. and S.L.: Methodology, Writing – review & editing.

*Competing interests.* Some authors are members of the editorial board of journal The Cryosphere.

*Acknowledgements.* M.I. and S.R.H. were funded by the Dutch Research Council (NWO) under grant no. *ALWGO.2018.043* and *OCENW.GROOT.2019.091*, respectively.



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
