# Peer review of "Brief Communications: Tides and Damage as Drivers of Lake Drainages on Shackleton Ice Shelf"

_EGUsphere, 2024_

## Referee Comment (RC1)

The paper by Sommer et al. presents an interesting analysis of lake drainage on the Shackleton Ice Shelf, alongside ice shelf damage and the evolution of tidal cycles. The authors draw several conclusions about the predominant regions of lake formation and drainage, based on the degree of damage and activity of this variable. The methods used in this paper are well described and defined in previous studies, particularly for drainage detection and damage assessment, and thus represent an interesting methodological application.

In general, this paper is clear and well-structured. The authors also clearly and extensively present the limitations of the methodology used. However, there are several aspects that need further elaboration to clarify the conclusions, particularly regarding the methodology and the results derived from it. Indeed, the current version of the paper could be more convincing, especially in highlighting the links between drainage, damage, and tidal cycles (see general comments below).

These comments should be addressed before the publication of this study.
* * *
**General Comments:**

An important point to clarify is the definition of damage. Initially, damage is defined in modeling as a variable of the enhancement factor, which modulates the ice's fluidity. This modeling variable is defined between 0 and 1 and can be adjusted to best fit the observed ice flow. Here, the authors completely recalculate this damage variable from satellite observations, independently of a model or flow velocities. It is not entirely clear how these products could be directly used in a flow model. Some studies have suggested that the crevasses derived by NerD do not match the damage modeled in an ice flow model (Gerli et al., 2024). Therefore, to avoid confusion between the terminology used by modelers and that of this paper, I suggest using the term "satellite-derived damage" or something similar throughout the manuscript.

Regarding the calculation of damage, why didn't the authors use the same optical images as for lake detection? It seems that NerD also works with this type of image. The paper lacks details on the time coverage of Sentinel-1 vs. Sentinel-2 images. Do the dates align perfectly? What is the time delay? There are also unclear areas regarding damage calculation: only one map is presented—is it a mosaic? Over what period was it calculated? Why don't the authors present a time series of damage alongside the time series of lake drainage? This would better specify the exact timing of events, particularly concerning tidal cycles: for instance, does damage increase with tides? Does this increase precede drainage events?

A second point concerns the features detectable via the NerD algorithm. Looking at Figure 1A, it seems the algorithm effectively detects dislocation zones with wide surface crevasses, as observed near Denman's grounding line or north of Denman Ice Shelf. However, examination of a Landsat image (see below) reveals that almost the entire Denman Ice Shelf is heavily fractured by basal crevasses, unlike the Shackleton Ice Shelf. Yet, damage maps show very similar low damage values, suggesting an underestimation of basal fracturing. Given this limitation, I think the interpretation of results should be revisited, and this should be mentioned in the discussion, specifically in the sense that ice shelf can still be damaged from below, if it is not detected by your method.

Additionally, a significant portion of the Denman Ice Shelf's front appears to be detached from Shackleton, more so than shown in Figures 1 and 2. Thus, some of the damage signal may no longer belong to the ice shelf and could merely represent calving icebergs. Analyzing lake evolution in this sector is therefore limited. The Shackleton Ice Shelf's frontal boundaries are also ambiguous (see image below), with a border region between sea ice and mixed ice that could bias the damage analysis. How are you dealing with these regions?

[Figure]

*L200-215:* How does the coarser resolution affect your interpretation? Drainage is a highly localized phenomenon. When downsampling, you might "leak" damage values located far from drainage events, especially with a factor of 10. Why was a factor of 10 used? Were smaller downsampling values tested unsuccessfully? If so, this could indicate that the correlation between damage and lake drainage events is not as strong as suggested, and tuning the data by downsampling may not be the correct approach?

Furthermore, most figures shown in this paper are very general and synthetic, for the sake of the "brief communication" format. This significantly limits the analysis of results. For example, it is important to include a zoomed-in or more detailed figure focusing on a lake drainage event that clearly shows the relationship with damage/crevassed regions (or a new panel of Fig 2). Similarly, it would be important to include a Sentinel-1 image with the retrieved damage (at least in an appendix) and the visually observed fracturing. I am unsure of the best solution: either move the paper to the regular format of *The Cryosphere* or add one or two figures to the appendix.

Concerning the tidal analysis, the authors argue that drainage events always occur during the ascending phase of tides. Looking at Figure 3, I would argue this is a bit of a stretch. Indeed, 6/11 drainage events in 2019 (more than half) started during the lowest (or even descending) phase of the tidal cycle (drainages M, L, F, H, J, E). The same applies to 2020, which saw only one drainage event. This does not undermine the paper's conclusions, as it remains plausible that changes in stress on the ice shelf with tidal cycle, and thus damage, favour these drainage events. For example, could the authors argue that the descending phase of the tidal cycle might be even more prone to crevasse opening due to flexure, whereas the ascending phase might favor crevasse closure?
* * *
**Specific Comments:**

- **L46:** Does lake advection with ice flow affect your mosaic calculation in any way?
- **L65:** Can you justify the choice of 80% loss of area?
- **L67-70:** Can you justify the threshold choices? Why 54,000 m²? Why a median lake depth greater than 0.65 m?
- **L84:** Why don't you consider rifting? Rifts could also be an important source of lake drainage.
- **L115:** Same as before, better justify the threshold used to classify damage.
- **L127:** If the activeness value mainly reflects shearing, why not directly use shear strain rates? What is the added value of activeness?
- **L137:** What do you mean by "more pronounced"?
- **L150-155:** Can you clarify how you can have low damage but high activeness?
- **L195:** Do we really need more data and satellites? With Sentinel-2 providing data every 5 days, Sentinel-1 every 6–12 days, and Landsat-8-9 every 15 days, what is your temporal sampling of drainage events when combining these data? What is the finest temporal scale you could achieve?

**Figures:**

- **Fig1:** Meltwater extent appears highly correlated with Denman's shear margins. Has any analysis been conducted in this regard?
- **Fig2:** What do you mean by pixel quality? Be more specific. This figure also needs a close-up on an actual drainage event, showing the related fracturing observed in the raw data (see earlier comment).
- **Fig3:** Include damage evolution with the tidal cycle (see earlier comment).

---

## Referee Comment (RC3)

**Review of: Tides and Damage as Drivers of Lake Drainages on Shackleton Ice Shelf**

January 2025

**1 General**

The goal of the work is very straightforward and well defined. Basically, there are three main goals: 1) map the overlap of damage and ponds; 2) find the draining ponds and identify in which conditions they occur; and 3) identify triggering events.

It is an interesting study, very important in the study of Antarctica. However, it is hard to define how impactful the findings are, since the methodology lacks more description of the metrics. Furthermore, the analysis should be deeper (see Major Comments).

I think the manuscript has 4 main issues:

1. observations do not support the claims in the manuscript

2. lack of a more robust statistical analysis

3. small number of observations

4. more descriptive methodology and how are the thresholds defined

**2 Major Comments**

**2.1 Sample size**

The authors used a time span of 3 years for the analysis performed. They found 13 events, which is a small sample size. Why do you not cover a larger time span to have more data? In addition, covering other regions would be beneficial. Furthermore, the definition of threshold values seems quite arbitrary, indicating that they fit only to these present specific conditions.

Also, the authors highlight the need for more sophisticated statistical approaches. This is a major concern for me, because not only the dataset is small, but it also lacks meaningful statistical analysis.

**2.2 Not supportive data**

I think the data presented do not support the conclusions drawn. In fact, the authors repeatedly make an affirmation and draw it back after a few sentences, for example:

L139-140 = "However, all of the detected lake drainage events occur in areas of the ice shelf classified as medium to highly active"

and

L151-L153 = "For example, drainages H, K, and M took place in areas with relatively low damage but high activeness, while drainages A, F, and G occurred in the least active regions, yet showed high levels of damage."

Furthermore, analysis of Figure B1 reveals that 1 of the events is in a low activeness area, undermining the claim in L139-140.

**2.2.1 Activeness parameter**

The authors conclude that "all of the detected lake drainage events occur in areas of the ice shelf classified as medium to highly active", but they sum up to 90% of the studied region. Taking into account that they have only 13 drainage events, their conclusion is not supported by the data since further analysis should be made.

Even if drainage events are correlated to activeness, it could be the case that activeness is actually related to damage (meaning that damage is higher when it is caused by the flow, which is very plausible) and that drainage events are mainly driven by damage (also very plausible).

**2.2.2 Section "Lake Drainage Events in Periods of Increasing Tidal Heights"**

As reviewer #1 said "6/11 drainage events in 2019 (more than half) started during the lowest (or even descending) phase of the tidal cycle (drainages M, L, F, H, J, E)". This simply invalidates the sentence: "Our findings unveil a compelling narrative of ice shelf dynamics, revealing an intricate interplay between tidal forces and supraglacial lake drainage events".

Furthermore, in a hypothetical case where all the drainage events occur in an ascending-amplitude phase of the tidal, it will not imply what is said: if you have thousands of lakes in a ice shelf, and each time only a small fraction of lakes drain, the likelihood of a drainage event triggered by tidal flexure would be higher in the crest of the amplitude phase.

**3 Minor comments**

**3.1 Methodology clarification**

Some clarifications are needed on the two main metrics used in the study. First, I agree with reviewer #1 about the lack of information on how damage is calculated. Usually it is inverted from

$$\mu = \frac{(1-D)B}{2\dot{\varepsilon}_e^{\frac{n-1}{n}}} \tag{1}$$

where $\mu$ is the ice viscosity, $D$ is damage (which you want to invert), $B$ is the ice rigidity, $\varepsilon_e$ is the effective strain rate, and $n$ is the flow law exponent. I think it is needed to specify how damage is calculated from remote sensing. Also, if possible, it would be interesting to relate the damage calculated in the present work (which reviewer #1 suggests changing the name to "satellite-derived damage" and I agree), and damage calculated from Equation 1.

**3.2 Activeness vs. damage**

I think a further analysis of the relationship between damage and activeness is required. It can be the case that the relationship between them is high, so any relationship between activeness and drainage occurrences is only due to damage.

**3.3 Motivation**

The first sentence of the article ("Surface lake drainage can destabilize ice shelves, occurring either slowly via supraglacial channels or rapidly through crevasses") makes me wonder if you are not studying the opposite. Instead of analyzing how damage influences lake drainage, should not you analyze how lake drainage influences damage? Otherwise, if you really want to analyze how damage influences lake drainage, you should motivate that in the introduction. I think you are putting the cart before the horse.

**4 Specific comments**

- L16: Is it the first time that "activeness" is used? If so, say that the manuscript introduce this concept. Otherwise, make a reference.

- L44: Do you advect the features when you merge the images in the mosaic?

- L50: This sampling frequency (once a year) is very different from the optical image that you are going to contrast later on. How do you deal with that?

- L52: It is not clear where you use the velocity field. For sure for the "activeness" calculation, but do you also use to transport the features? Make it clear near the description of the velocity field calculation.

- L61: Regarding the threshold 1800 m$^2$ you make a citation for this value, but what is the reasoning of using this threshold?

- L65: Why 80%? I can also imagine 50% as a massive drainage event. this looks like a random choice, that you need to pick one, but if you lower the threshold, you would have many more drainage events, increasing the data you can use to infer, since 13 drainage events are not many.

- L67: Was not the are threshold 1800? Furthermore, why do you use these threshold? Give a reason and use the citation. Only the citation is not enough.

- L72: you excluded 20 out of 25, so you have 5 drainage events. How then you have 13 drainage events in your results?

- L78: Add "resolution" after "300 m".

- L83: Missing citation. Makes sense to compare the angle of the fracture to its orientation, but quantifying it can be tricky. Is there any supporting studies for the use of that values?

- L88: If activeness is binary, how you produce an image like Figure 2 b)? It do not seems like a product of downsampling.

- L88: I don't see the reason of donwsampling it.

- L89: Why normalizing? You can say directly that damage varies between 0 and 1 and everything is already normalized.

- L89: Add "resolution" after "3000 m".

- L101: Add "total" in "with total maxima".

- L116: I think the definition of the thresholds should go to methods with a further explanation of the threshold used.

- L119-124: This is a very sounding result, supporting the hypothesis of a strong relationship between drainage events and damage. However, I would expect the same analysis regarding the activeness. The distribution of 10%, 71%, and 19% does not allow this analysis. Contrasting the area distribution and Figure B1, and do not see a strong relationship between activeness and drainage events. I would say that most of the signal of drainage events are due to damage.

- L129: Same comment as for damage.

- L139: This conclusion is not surpirse. It sums 90% of the studied area.

- L151: What do you mean by "parallel trend". Be more specific.

- L151-153: This goes against the phrase: "However, all of the detected lake drainage events occur in areas of the ice shelf classified as medium to highly active".

- L156: You previously said the opposite.

- L158: As far as I understood, it is impossible to drainage event to occur without damage. Be more precise with this statement.

- L161: I agree with reviewer #1 regarding the drainage events with respect to the ascending phase of tidal cycles.

- L187: I think you do not have drainage events that last hours. If this is the case, remove the "few hours".

- L190: Here you say that it is difficult to assign the drainage events to hydrofracturing, but in the discussion you did this.

- L197: If NeRD can not identify individual fractures, how then you measure the orientation of the fractures to calculate the activeness?

- 214-216: This sentence is another major concern for this study.

**4.1  Figures and tables**

I would appreciate a hexabin graph with activeness and damage in the axes. This would allow us to see the correlation between both metrics and the occurrence of drainage events.

- Table A1: Bring it to the main body of the text, it is too important. I suggest including two more columns: Classification of damage and activeness (low, medium, high).

- Figure 1: Define LIMA as an optical imagery mosaic from Landsat.

- Figure 2: Add ", respectively" at the end of the first sentence.

---

## Author Comment (AC1)

**Reviewer 1**

We thank the reviewer for their time and feedback on the manuscript. The valuable comments and questions are carefully considered, and we discuss below how we intent to incorporate their suggestions in the next version of the manuscript.

In short (also considering comments from the other reviewers), we will focus on (a) extending our temporal analysis to increase our sample size of observed lake drainages, (b) extend analysis of our activeness parameter by comparing to the vulnerability map of Lai et al. (2020) and to the strain rates (c) provide more details on methods, specifically the NeRD algorithm and the definition of thresholds and (d) will implement major textual changes to better clarify and align our conclusions to our observations.

For some (minor) edits we have already started implementing changes, and for some comments we can already provide some provisional additional figures in this document. We thank you for understanding this was not feasible yet for all comments.

**General Comments:**

*An important point to clarify is the definition of damage. Initially, damage is defined in modeling as a variable of the enhancement factor, which modulates the ice's fluidity. This modeling variable is defined between 0 and 1 and can be adjusted to best fit the observed ice flow. Here, the authors completely recalculate this damage variable from satellite observations, independently of a model or flow velocities. It is not entirely clear how these products could be directly used in a flow model. Some studies have suggested that the crevasses derived by NerD do not match the damage modeled in an ice flow model (Gerli et al., 2024). Therefore, to avoid confusion between the terminology used by modelers and that of this paper, I suggest using the term "satellite-derived damage" or something similar throughout the manuscript.*

Indeed, as the reviewer states, our use of 'damage' is different from the definition in damage mechanics models. It is also not a parameterisation that can directly be used in flow models since there is no quantified translation from damage signal to fracture depth (as is explained in Izeboud and Lhermitte (2023)). We will more explicitly define 'damage' in the manuscript as 'visible features of damage at the ice shelf surface' as an umbrella for crevasses, rifts and fractures.

We also appreciate the suggested term, and will adjust the manuscript to use 'satellite-derived damage' or "damage signal" when discussing our observations.

*Regarding the calculation of damage, why didn't the authors use the same optical images as for lake detection? It seems that NerD also works with this type of image. The paper lacks details on the time coverage of Sentinel-1 vs. Sentinel-2 images. Do the dates align perfectly? What is the time delay? There are also unclear areas regarding damage calculation: only one map is presented — is it a mosaic? Over what period was it calculated? Why don't the authors present a time series of damage alongside the time series of lake drainage? This would better specify the exact timing of events, particularly concerning tidal cycles: for instance, does damage increase with tides? Does this increase precede drainage events?*

While NeRD is compatible with both optical and radar imagery, we opted for Sentinel-1 radar data for damage detection. This choice was driven by the fact that optical images, including those from Sentinel-2, are affected by cloud cover, limiting their usability for consistent monitoring. In contrast, Sentinel-1 provides reliable coverage over short time periods, ensuring a more complete dataset for our analysis.

Regarding the damage map, we acknowledge that our methods section could have been more explicit. We created a single damage map by selecting individual Sentinel-1 images from September to November of each year that cover the Shackleton Ice Shelf. Each image was processed using NeRD, and we calculated the median damage value from the image stack. However, due to variations in satellite coverage, not all areas of the ice shelf were imaged with the same frequency.

The time period for damage assessment does not specifically overlap with the lake drainage events. This is because NeRD detects linear features based on grayscale contrasts and is sensitive to edges between water and snow. To minimize potential interference from surface meltwater, we used images from before the melt season, which also reinforces the use of Sentinel-1 over Sentinel-2 for this purpose.

In terms of damage characterization, NeRD is designed to detect large-scale damage features (>100 m) and assess the spatial extent and growth of damaged areas. However, it is not suited for tracking fine-scale crevasse development or deepening. While investigating potential correlations between crevasse changes, tidal phases, and drainage events would be highly valuable, it is not feasible given the resolution and capabilities of our current dataset.

To enhance transparency and provide additional insights, we will include damage maps for each season in the supplementary materials. We hope these additions, along with the clarifications above, address your concerns.

*A second point concerns the features detectable via the NerD algorithm. Looking at Figure 1A, it seems the algorithm effectively detects dislocation zones with wide surface crevasses, as observed near Denman's grounding line or north of Denman Ice Shelf. However, examination of a Landsat image (see below) reveals that almost the entire Denman Ice Shelf is heavily fractured by basal crevasses, unlike the Shackleton Ice Shelf. Yet, damage maps show very similar low damage values, suggesting an underestimation of basal fracturing. Given this limitation, I think the interpretation of results should be revisited, and this should be mentioned in the discussion, specifically in the sense that ice shelf can still be damaged from below, if it is not detected by your method. Additionally, a significant portion of the Denman Ice Shelf's front appears to be detached from Shackleton, more so than shown in Figures 1 and 2. Thus, some of the damage signal may no longer belong to the ice shelf and could merely represent calving icebergs. Analyzing lake evolution in this sector is therefore limited. The Shackleton Ice Shelf's frontal boundaries are also ambiguous (see image below), with a border region between sea ice and mixed ice that could bias the damage analysis. How are you dealing with these regions?*

We acknowledge the limitation that NeRD tends to underestimate basal fracturing, as basal crevasses are often represented as low damage values or may not be detected at all. We will explicitly clarify this in the discussion to ensure a more nuanced interpretation of the results, emphasizing that ice shelves can still experience significant damage from below, even if it is not captured by our method.

Regarding the Denman Ice Shelf front, we appreciate this important consideration. The damage maps were clipped to ice shelf fronts that were manually delineated based on Sentinel-1 observations from September to November. However, the impact of any inaccuracies in the ice shelf front positioning is limited due to the following factors:

(a) The velocity maps were masked more strictly than the damage maps, effectively removing the outermost portion of the Denman ice front, as seen in Figure 2.

 (b) Only one drainage event (event F) was identified in that region, and all detected drainage events were manually verified to exclude false positives.

We also recognize the ambiguity in the Shackleton Ice Shelf's frontal boundaries and the potential influence of mixed sea ice in these regions. We will clarify how these areas were handled in our analysis and ensure that any potential biases in the damage assessment are acknowledged.

We appreciate your detailed feedback and will incorporate these points into the revised manuscript.

*L200-215: How does the coarser resolution affect your interpretation? Drainage is a highly localized phenomenon. When downsampling, you might "leak" damage values located far from drainage events, especially with a factor of 10. Why was a factor of 10 used? Were smaller downsampling values tested unsuccessfully? If so, this could indicate that the correlation between damage and lake drainage events is not as strong as suggested, and tuning the data by downsampling may not be the correct approach? Furthermore, most figures shown in this paper are very general and synthetic, for the sake of the "brief communication" format. This significantly limits the analysis of results. For example, it is important to include a zoomed-in or more detailed figure focusing on a lake drainage event that clearly shows the relationship with damage/crevassed regions (or a new panel of Fig 2). Similarly, it would be important to include a Sentinel-1 image with the retrieved damage (at least in an appendix) and the visually observed fracturing. I am unsure of the best solution: either move the paper to the regular format of The Cryosphere or add one or two figures to the appendix.*

We will add a zoomed-in figure of a lake drainage event and its relationship with damage/crevassed regions, either as an additional panel in Figure 2 or as a supplementary figure. Additionally, a Sentinel-1 image with the retrieved damage and visually observed fracturing will be included in the supplementary material.

Regarding downsampling, NeRD detects broader damaged areas rather than individual crevasses, meaning even at 300 m resolution, the damage maps do not pinpoint exact crevasse locations. We hypothesize that overall ice weakness, rather than specific crevasse presence, is the key factor influencing lake drainage. The activeness parameter is designed to capture this broader structural weakness.
We tested different downsampling approaches and found that correlating lake drainages with damage remains difficult due to the widespread nature of damage and the limited number of observed drainage events. This challenge persists regardless of resolution, indicating that while a relationship likely exists, it is not strongly defined.

*Concerning the tidal analysis, the authors argue that drainage events always occur during the ascending phase of tides. Looking at Figure 3, I would argue this is a bit of a stretch. Indeed, 6/11 drainage events in 2019 (more than half) started during the lowest (or even descending) phase of the tidal cycle (drainages M, L, F, H, J, E). The same applies to 2020, which saw only one drainage event. This does not undermine the paper's conclusions, as it remains plausible that changes in stress on the ice shelf with tidal cycle, and thus damage, favour these drainage events. For example, could the authors argue that the descending phase of the tidal cycle might be even more prone to crevasse opening due to flexure, whereas the ascending phase might favor crevasse closure?*

We understand the concern. We would like to clarify that indeed drainage events M, E and K 'start' in the descending phase (3/13) and 4/13 in the lowest phase. However, with this, it is important to realise that the drainage events shown in figure 3 occur between the detected dates t1 and t2, since we only detect 'lake is present' at t1 and 'lake has drained' at t2 – it is not a draining that starts at t1 and ends at t3. Therefore, Figure 3 does (in our opinion) clearly suggest that drainage does not occur in the descending phase (even though, indeed, crevasse might be more prone to opening in that phase!). We do acknowledge that we cannot differentiate if the drainage occurs in the lowest part of the cycle or the ascending part, so will adjust the text accordingly.

**Specific Comments:**

*L46: Does lake advection with ice flow affect your mosaic calculation in any way?*
No, this effect is negligible. The mosaics are calculated on a very short period of time (8-10 days) and thus the only area where advection would be significant is at the fast flowing Denman Glacier (max speeds of ~1500 m/yr (Miles et al. 2020[1]), ~4.6 m/day, just enough to have 1 pixel of advection in the 30 m Sentinel-2 images in the selected period) -- ice flow speed quickly drops to <= 500 m/year to the sides of the ice
* * *
[1] https://doi.org/10.5194/tc-15-663-2021

tongue. Moreover, in this short period of time the amount of repeat satellite overpasses is very limited, and we get just the minimum amount of image to stitch them together for a domain-covering mosaic, thus having almost no impact of advection on the mosaic calculation). We'll include a supplementary figure in the next manuscript to show the overlap of the used satellite images to construct each mosaic.

*L65: Can you justify the choice of 80% loss of area?*
This is in line with other literature, following Doyle et al.(2013), Fitzpatrick et al. (2014), Miles et al. (2017), Williamson et al. (2017). These references will be specified in the text.

- *Doyle, S. H., Hubbard, A. L., Dow, C. F., Jones, G. A., Fitzpatrick, A., Gusmeroli, A., Kulessa, B., Lindback, K., Pettersson, R., and Box, J. E.: Ice tectonic deformation during the rapid in situ drainage of a supraglacial lake on the Greenland Ice Sheet, The Cryosphere, 7, 129–140, https://doi.org/10.5194/tc-7-129-2013, 2013*
- *Fitzpatrick, A. A. W., Hubbard, A. L., Box, J. E., Quincey, D. J., van As, D., Mikkelsen, A. P. B., Doyle, S. H., Dow, C. F., Hasholt, B., and Jones, G. A.: A decade (2002–2012) of supraglacial lake volume estimates across Russell Glacier, West Greenland, The Cryosphere, 8, 107–121, https://doi.org/10.5194/tc-8-107-2014, 2014*
- *Miles, K. E., Willis, I. C., Benedek, C. L., Williamson, A. G., and Tedesco, M.: Toward monitoring surface and subsurface lakes on the Greenland Ice Sheet using Sentinel-1 SAR and Landsat 8 OLI imagery, Front. Earth Sci., 5, 1–17, https://doi.org/10.3389/feart.2017.00058, 2017*
- *Williamson, A. G., Arnold, N. S., Banwell, A. F., and Willis, I. C. (2017). A Fully Automated Supraglacial lake area and volume Tracking ("FAST") algorithm: development and application using MODIS imagery of West Greenland. Remote Sens. Environ. 196, 113–133. doi: 10.1016/j.rse.2017.04.032*

*L67-70: Can you justify the threshold choices? Why 54,000 m²? Why a median lake depth greater than 0.65 m?*
Similarly as previous comment, these threshold choices for lake area and depth were based on previous studies that utilized similar criteria to define lakes of significant size and impact.Specifically, an area of 54,000 m² was chosen as it corresponds to 60 pixels in Landsat 8 imagery, providing sufficient resolution to capture lake characteristics. This area threshold is consistent with previous research (Williamson et al. (2018a), which considered lakes of this size to be relevant in terms of water volume and potential impact on ice shelves.*Williamson, A. G.; Banwell, A. F.; Willis, I. C.; Arnold, N. S. Dual-Satellite (Sentinel-2 and Landsat 8) Remote Sensing of Supraglacial Lakes in Greenland. The Cryosphere 2018, 12 (9), 3045–3065. https://doi.org/10.5194/tc-12-3045-2018 a.*

Regarding the median lake depth threshold of 0.65 m, this value was selected to filter out large but shallow areas that do not have a well-defined depth profile. The standard deviation of 0.3 m helps ensure that only lakes with substantial depth variations are included. These values are comparable to those used in other studies, such as those by Williamson et al. (2018b) on supraglacial lakes in Greenland.

*Williamson, A. G.; Willis, I. C.; Arnold, N. S.; Banwell, A. F. Controls on Rapid Supraglacial Lake Drainage in West Greenland: An Exploratory Data Analysis Approach. Journal of Glaciology 2018, 64 (244), 208–226. https://doi.org/10.1017/jog.2018.8 b*

*L84: Why don't you consider rifting? Rifts could also be an important source of lake drainage.*

We acknowledge that rifts can indeed be a source of meltwater drainage.

However, rifts typically form towards the front of the ice shelf and (by definition) extend through the entire thickness of the ice, which makes (nearby) water accumulation less likely. Instead, rifts are primarily locations of severe damage where water is more likely to run off directly rather than accumulate and form lakes. This aligns with our conclusions that while rifts can contribute to drainage, they are not conducive to lake formation. Nevertheless, indeed the widening or propagation of existing rifts could lead to lake drainages. This process, however, is not captured by the produced damage maps, since the NeRD algorithm does not detect individual fracture features.

Although distinguishing between rifts and crevasses can be challenging, our focus remains on understanding the broader dynamics of damage and its impact on lake drainage, without specifically isolating rifts as a separate category.

*L115: Same as before, better justify the threshold used to classify damage.*

Understood, and we'll further clarify in the text. Different than the melt lake thresholds, there is limited previous research using similar damage detection approaches. As the damage signal is a measure of feature contrast in the satellite image, there is unfortunately not a quantitative translation to physical properties (such as crevasse depth or density). We have therefore discretized the damage signal values to obtain a data-based estimate of what our 'low', 'medium' and 'high' values were. Due to the strongly skewed data distribution (supplementary Figure B1), we discretized the damage signal values in bins of unequal width, containing progressively less data samples (damaged pixels) to favor the representation of high values: the 'low', 'medium' and 'high' classes contain respectively 62.5%, 25% and 12.5% of the samples. The widths of the bins were obtained by initially applying a quantile-based discretization with 8 equal-sized buckets, and then grouping the first buckets into one to yield a reasonable bin-range: the first 5 buckets together contain damage signals between [0, 0.01) (classified as 'low') - which is still quite a small range, compared to the 'medium' [0.01, 0.2) and 'high' [>0.2] classes.

A similar approach was done to discretize the Activeness parameter. As this parameter has a normal distributed data, we divided the buckets to favor both the low and high ends of the curve, yielding the class 'low' activeness with 12.5% of samples, 'medium' activeness with 75% of samples, and high activeness with 12.5% of samples.

The thresholds in the submitted script deviate a bit from the here mentioned (no effect on the classification). The here presented approach represents our initial though process. We will adjust the thresholds in the manuscript accordingly.

*L127: If the activeness value mainly reflects shearing, why not directly use shear strain rates? What is the added value of activeness?*

While shear strain rates provide valuable information about the stress environment within an ice shelf, they do not capture the behaviour of observed damage features. Damage development (crevasse or fracture opening) is strongly linked to strain rates, but they advect and rotate with the ice as well. As a result, observed damage features might be found in areas downstream of where they formed, and might have become 'passive'. The activeness parameter is aimed to account for this, using the observed orientation (angle with respect to flow) to infer if the detected features are in a position that favors crevasse opening.

We will add a clarification in the text where the activeness parameter is introduced:

*As detected damage features might have advected downstream of the area where they formed, we infer a measure of local 'activeness' of the features. The obtained damage orientation is used to identify areas with a likelihood of active damage development, by comparing the damage orientation to local ice flow angle to infer if the feature is in a position that favors crevasse opening*

*L137: What do you mean by "more pronounced"?*

To clarify the term "more pronounced," we are referring to regions within glacier zones where ice flow velocities are particularly high. These areas are predominantly influenced by the strain and shear forces generated by the flowing ice. This dynamic environment is characterized by significant stress from adjacent slower-moving or stationary ice masses, which enhances the activeness of these regions.

**New sentence.** This pattern occurs where the moving ice experiences shear stress from adjacent slower-moving or stationary ice masses, leading to mix-mode crevasse opening (Colgan et al. 2016). Glacier zones, where ice flow is most pronounced, serve as the clearest examples of areas exhibiting high activeness Glacier zones, characterized by high ice flow velocities, serve as the most illustrative examples of areas exhibiting high activeness.

*L150-155: Can you clarify how you can have low damage but high activeness?*

In essence, high activeness indicates the potential for further damage development due to the local stress environment, while low damage simply reflects the current state of structural integrity. You can find small yet opening crevasses as well as large but stationary rifts. This distinction highlights that activeness is more about the dynamics and stress patterns in the ice shelf rather than the extent of existing damage.

*L195: Do we really need more data and satellites? With Sentinel-2 providing data every 5 days, Sentinel-1 every 6–12 days, and Landsat-8-9 every 15 days, what is your temporal sampling of drainage events when combining these data? What is the finest temporal scale you could achieve?*

The reviewer makes a good point, as more is not always better, and we might instead focus on a smarter use of the data that is available. In any case, hydrofracturing or rapid lake drainages occur on very short time scales (some even under 24 h) and being able to sample with an interval of 2 to 6 days would greatly benefit studying hydrofracturing processes. In our case, we utilized mosaics with a period of 8 to 10 days, which represents the finest possible temporal resolution we could achieve due to the relatively high cloud coverage during the Antarctic summer. For other regions with less cloud cover a finer temporal resolution can probably already be achieved, or by integrating radar (Sentinel-1) in the detection approach: e.g. in West Greenland, Miles et al. (2017) achieved a 3 day interval by combining Sentinel-1 and Landsat-8.

We'll adjust the text to give more context: "..., rapid filling and drainage events that occur entirely between image acquisitions may go undetected completely; cloud cover and other atmospheric disturbances can obscure satellite imagery and potentially mask drainage events that occur on timescales of 2-6 days (Miles et al., 2017). The limitation of using 8-10 day mosaics underscores the need for higher temporal resolution in satellite data or the integration of complementary observational methods to accurately identify and attribute rapid drainage events and hydrofracture mechanisms."

*Fig1: Meltwater extent appears highly correlated with Denman's shear margins. Has any analysis been conducted in this regard?*

Figure 1 displays the extent of meltwater lakes, which is different than the total meltwater extent on the ice shelf. On the Shackleton ice shelf it is not uncommon that the whole domain experiences melt at some point in season (see also De Roda Husman et al., 2024 and Saunderson et al., 2022). The meltwater lakes accumulate in areas depending on surface topography, firn air content and the amount of melt, and it is not strange to see this in the shear margins (i.e. lots of available surface depressions for water accumulation due to ice deformation). Our observations are consistent with the observed lakes by Arthur et al. (2020) on the eastern half of the Denman glacier, included below for clarity.

- *de Roda Husman, S., Lhermitte, S., Bolibar, J., Izeboud, M., Hu, Z., Shukla, S., van der Meer, M., Long, D., and Wouters, B.: A high-resolution record of surface melt on Antarctic ice shelves using multi-source remote sensing data and deep learning, Remote Sensing of Environment, 301, 113 950, https://doi.org/https://doi.org/10.1016/j.rse.2023.113950, 2024.*
- *Saunderson, D., Mackintosh, A., McCormack, F., Jones, R. S., and Picard, G.: Surface melt on the Shackleton Ice Shelf, East Antarctica (2003–2021), The Cryosphere, 16, 4553–4569, https://doi.org/10.5194/tc-16-4553-2022, 2022*

- *Arthur, J. F., Stokes, C. R., Jamieson, S. S. R., Carr, J. R., and Leeson, A. A.: Distribution and Seasonal Evolution of Supraglacial Lakes on* Shackleton Ice Shelf, East Antarctica, The Cryosphere, 14, 4103–4120, https://doi.org/10.5194/tc-14-4103-2020, 2020

[Figure]

*Figure from Arthur et al. (2020) of detected surface meltwater lakes on a subset area of the Shackleton Ice Shelf.*

*Fig2: What do you mean by pixel quality? Be more specific. This figure also needs a close-up on an actual drainage event, showing the related fracturing observed in the raw data (see earlier comment).*
We will rephrase the caption.

**Figure 2.** Spatial occurrence of lake drainages on Shackleton ice shelf. a) and b) present NeRD-derived damage and activeness metrics for the 2019/20 melt season, as derived in subsection 2.4 respectively. The bottom row shows a zoomed section around the glaciers on the ice shelf, with c) NeRD damage d) maximum lake extent, and e) activeness for the melt season 2019/20. Coloured dots with blue labels: Drainage events detected during the Antarctic melt seasons of 2018/19, 2019/20 and 2020/21. The colour of each dot represents either activeness or damage at its location for the respective melt season in which it occurred. . Thick black line: Grounding line. Thin black line: Shelf coastline. Both from from MEaSURES data set by Gerrish et al. (2022).

And indeed, we will provide close ups of the drainage events. An example is shown below. The figure below shows the S2 timeseries of a detected lake (E as in figure 2) that is detected to be draining between 2020-01-30 and 2020-02-13. We will also add S1 and L8 in supplementary, based on which we have narrowed down the draining window.

[Figure]

*Fig3: Include damage evolution with the tidal cycle (see earlier comment).*

Thank you, we do agree that studying the damage evolution with the tidal movement would be a valuable addition. As also stated in the earlier comment, it is not straightforward to include the desired damage evolution from NeRD in this way. To mitigate potential misinterpretation of meltwater as damage by NeRD, we have assessed damage before each melt season (in September). Moreover, given Nerd's nature of utilizing a 10-pixel window size and subsequent loss of spatial resolution, it is questionable whether NeRD is the appropriate tool for finding meaningful temporal changes in damage with a temporal resolution that matches tidal movements. Instead, NeRD is better suited for investigating damage patterns.

---

## Author Comment (AC2)

**Reviewer 2**

We thank the reviewer for their time and feedback on the manuscript. The valuable comments and questions are carefully considered, and we discuss below how we intent to incorporate their suggestions in the next version of the manuscript.

In short (also considering comments from the other reviewers), we will focus on (a) extending our temporal analysis to increase our sample size of observed lake drainages, (b) extend analysis of our activeness parameter by comparing to the vulnerability map of Lai et al. (2020) and to the strain rates (c) provide more details on methods, specifically the NeRD algorithm and the definition of thresholds and (d) will implement major textual changes to better clarify and align our conclusions to our observations.

For some (minor) edits we have already started implementing changes, and for some comments we can already provide some provisional additional figures in this document. We thank you for understanding this was not feasible yet for all comments.

**General Points**

One issue is that the study is quite limited in scope, both in terms of space and time. The generalizability of the conclusions is very limited by the fact that this study only covers three years and one relatively small ice shelf. Given the computing platform the authors used (google earth engine), it seems like it could have been relatively simple to extend the work to other locations and, within the restrictions of the datasets used, to more melt seasons.

The main novelty of the paper is the consideration of 'activeness' (i.e. how perpendicular is ice flow to the predominant fracture orientation). And one could imagine a paper on this topic (1) proposing activeness as an important factor and (1) thoroughly test if this is the case. However, without an extension in the temporal and spatial coverage of the analysis, I think it is difficult to conclude much about the importance or otherwise of activeness for lake drainage. For example, as I note below, the authors state "…it is clear that at least one of these factors, damage or activeness, must be present for lake drainage to occur." (albeit with a caveat that the exact relationship needs further investigation). I think, given the results presented here, this is a much too strong conclusion to draw at this stage. That leaves the main contribution of the paper as (1) proposing activeness as an important factor. It is worth making it clearer that this is the main contribution of the paper. Or alternatively extending the analysis so that it can achieve (2) as well. This would involve extending to other ice shelves and/or other time periods.

We recognize the limitations of our study, and understand the concerns of the reviewer as a result of that. As the reviewer describes clearly below, the main aim of this study was to assess when/where the combination of melt lakes and damage would (not) lead to hydrofracturing, and we propose activeness as an important factor. We concur that some conclusions might be worded too strong and will rephrase the manuscript to focus on that contribution and bring more nuance to the conclusions; more specifics are explained below. Our intent was, and remains, to present a case study to address the factors contributing to hydrofracturing/lake drainages, which is also why we chose a Brief Communication format. We think that the Shackleton ice shelf presents an ideal case study with multiple 'types' of lakes and lake drainages, as well as various damage features. We agree that a larger scale application would be very valuable and will extent our analysis to the time period 2015-2024 (adding 6 more years, and hopefully many more drainage events).

Other important potential extensions that are missing include between lake drainage locations and ice-shelf stresses and/or vulnerability to hydrofracture as proposed by Lai et al. (2020). These are obliquely discussed in the context of introducing the idea that how close fractures are to perpendicular to ice flow (activeness) is an important factor. However, no comparisons are made to them directly. This would seem like a more direct way (or at the least a complementary way) of testing the ideas that lie behind the activeness proposal.

This is a valuable suggestion, and we'll incorporate a comparison to the vulnerable regions as detected by Lai et al. (2020) in the next version of our manuscript. A preliminary comparison can be seen in the (draft) figure below, showing their assessment at Shackleton, indicating the (non-)vulnerability to hydrofracturing of the ice shelf areas, i.e. if observed fractures are unstable if inundated with water. Our detected hydrofracturing events are superimposed on the map, with the color indicating the assessment of Lai et al. (2020) at their location, showing that 7 of the 13 events are in an area that they assessed as resistive to hydrofracturing; compared to all events occuring in areas of the ice shelf we identified as medium to highly active (seven and

six lake draining events respectively, Figure 2). We will expand on our results and discussion including this figure and provide similar comparison to strain rates.

[Figure]

**Specific comments**

*L14: "only some of these lakes drain." This needs citations*

We will add two references to support that statement:

Stokes, C. R.; Sanderson, J. E.; Miles, B. W. J.; Jamieson, S. S. R.; Leeson, A. A. Widespread Distribution of Supraglacial Lakes around the Margin of the East Antarctic Ice Sheet. *Sci Rep* 2019, *9* (1), 13823. https://doi.org/10.1038/s41598-019-50343-5.
Arthur, J. F.; Stokes, C. R.; Jamieson, S. S. R.; Rachel Carr, J.; Leeson, A. A.; Verjans, V. Large Interannual Variability in Supraglacial Lakes around East Antarctica. *Nat Commun* 2022, *13* (1), 1711. https://doi.org/10.1038/s41467-022-29385-3.

*L17: missing "an"*

Agreed and implemented.

*L44: reference should not be in parentheses*

Agreed and implemented.

*L44: explain what "Median image mosaics" are in more detail*

Thank you for your suggestion. We construct median image mosaics by selecting satellite images within the respective time period (10 or 8 days), filtering for cloud cover (<30%) and taking the pixel-wise median value of the image stack. In practice this means that for some parts of the ice shelf there's only a single overpass in the short time period and the "median" value of 1 sample is calculated. This short description will be added to L44.

*L46: comma before 'which'*

Agreed and implemented.

*Section 2.1: Inconsistent tense. Are and was are both used. Change to be consistent.*

Checked and Corrected.

*L48: not clear what grid is being referred to here. Is this a spatial reference grid? If so, give its name. Or maybe it is not needed here, unless it's important for the analysis that it was this grid for some reason.*

The grid mentioned here is a simple processing approach to tile the data in smaller files to streamline the processing of mosaics. It is not strictly necessary and will be removed for clarity.

*L52: rephrase "Antarctic ice flow velocity observations of 2019"*

We have changed the wording to:

Annual ice flow velocity data are sourced from the ITS_LIVE campaign (Gardner et al., 2020).

*L53: delete "and kept constant during the studied period."*

Agreed and implemented.

*Figure 1 caption: add 'the' before MEASURES.*

Agreed and implemented.

*L62: rephrase "combining all lake masks of a season."*

We hope with the following we can clarify the lake extent creation.

**L55:** The location and depth of supraglacial lakes are determined using a threshold-based method on the collected mosaics.

**L62:** For each melt season, the maximum lake extent is derived by aggregating all lake masks per mosaic generated during that period.

*L78: "damage maps of 300 m" should be "300m-resolution damage maps"*

Agreed and implemented.

*L83: I am unclear what "(tolerating a deviation of 15∘)" means.*

We have precised our answer and revised the text for clarity:

**L83:** Then, an area was considered likely to be 'active' if the difference between the damage orientation and the flow angle falls within 45° ± 15°, which occurs mainly in the shear zones of the ice shelf or in regions exhibiting mixed-mode opening fractures."

*L98: delete "under-"*

Agreed and implemented.

*Figure 2 caption: rephrase "show NeRD damage and activeness of summer 2019/20" and "pixel quality" (I am not sure what quality refers to here).*

Below is our revised caption:

**Figure 2.** Spatial occurrence of lake drainages on Shackleton ice shelf. a) and b) present NeRD-derived damage and activeness metrics for the 2019/20 melt season, as derived in subsection 2.4 respectively. The bottom row shows a zoomed section around the glaciers on the ice shelf, with c) NeRD damage d) maximum lake extent, and e) activeness for the melt season 2019/20. Coloured dots with blue labels: Drainage events detected during the Antarctic melt seasons of 2018/19, 2019/20 and 2020/21. The colour of each dot represents either activeness or damage at its location for the respective melt season in which it occurred. Thick black line: Grounding line. Thin black line: Shelf coastline. Both from from MEaSURES data set by Gerrish et al. (2022).

*L103: explain what "record summer" means explicitly. I am guessing it means record high temperatures and/or melt volume/extent. But which of these, I am not sure.*

You are right, the term "record summer" refers to the 2019-2020 season being notable for its unprecedented melt extent and duration, record surface meltwater ponding, and anomalously high air temperatures (Banwell et al., 2021). This will be edited in the manuscript.

L104: delete 'such as those by' and put the citations in parentheses
Agreed and implemented.

Agreed and implemented.

In the sentence in L100, "However, our findings indicate minimal lake formations in this region, suggesting immediate drainage of meltwater into the ocean in highly damaged areas" we mean the drainage/runoff of meltwater without first forming a meltwater lake. Therefore this is not included in Figure 3 (lake drainage events). We have added "..., without meltwater lake formation" to the sentence

*L105-107: "However, the findings indicate minimal lake formations in this region, suggesting immediate drainage of meltwater into the ocean in highly damaged areas (Figure 1 and 2)." This is not clear. Which part are you referring to as the north of the ice shelf? The north-west-most part is not covered by the damage map and the region to the south of that is not ubiquitously highly damaged, so I am not sure this is a fair conclusion to reach from these two maps (damage and meltwater).*
Our apologies, this was indeed unclear. In this statement, we meant the area north of Masson Island, we'll clarify in the text. Also, we will add the north tip of West Shackleton to complete our (damage) map in the next version of the manuscript.

*L116: how did you decide on these values for the categories no, medium and high?*

Understood, and we'll further clarify in the text. Different than the melt lake thresholds, there is limited previous research using similar damage detection approaches. As the damage signal is a measure of feature contrast in the satellite image, there is unfortunately not a quantitative translation to physical properties (such as crevasse depth or density). We have therefore discretized the damage signal values to obtain a data-based estimate of what our 'low', 'medium' and 'high' values were. Due to the strongly skewed data distribution (supplementary Figure B1), we discretized the damage signal values in bins of unequal width, containing progressively less data samples (damaged pixels) to favor the representation of high values: the 'low', 'medium' and 'high' classes contain respectively 62.5%, 25% and 12.5% of the samples. The widths of the bins were obtained by initially applying a quantile-based discretization with 8 equal-sized buckets, and then grouping the first buckets into one to yield a reasonable bin-range: the first 5 buckets together contain damage signals between [0, 0.01) (classified as 'low') - which is still quite a small range, compared to the 'medium' [0.01, 0.2) and 'high' [>0.2] classes.
A similar approach was done to discretize the Activeness parameter. As this parameter has a normal distributed data, we divided the buckets to favor both the low and high ends of the curve, yielding the class 'low' activeness with 12.5% of samples, 'medium' activeness with 75% of samples, and high activeness with 12.5% of samples.
The thresholds in the submitted script deviate a bit from the here mentioned (no effect on the classification). The here presented approach represents our initial though process. We will adjust the thresholds in the manuscript accordingly.

*Figure B2: I think this would be useful to have in the main paper.*
Thank you for the suggestion. We agree that Figure B2 provides valuable detail; however the Brief Communications format only allows for three display items (tables/figures) so we are very limited in our flexibility here, unfortunately.

*L118: replace "a detailed listing" with "a list"*
Agreed and implemented.

*L127: this is not an indication of where the ice is deforming, only of where it is flowing perpendicular to fractures. The ice is actively deforming essentially everywhere.*

You are indeed correct, deformation occurs everywhere on the ice shelf. This activeness metric specifically targets areas where the relative orientation between flow and fractures indicates dynamic activity with respect to the fracture only. We rephrased:

**L127:** The activeness parameter provides insights into the dynamic behavior of the ice shelf, with high values indicating areas where the local fractures have a high likelihood to be under active development due to the flow of the ice.

*L128: replace "distributed in a bell-shaped curve" with "normally distributed"*
Agreed and implemented.

*L131: clarify what "the glaciers," refers to here.*
We referred here to the main glaciers of Shackleton Ice Shelf. We refined the text by mentioning them:

**L131:** Regions with high activeness are observed around the glaciers Northcliffe, Denman, Scott and Apfel, as well as around the northern tip of West Shackleton, and along the grounding lines of both West and East Shackleton, refer to Figure 1 and 2b.

*L132: delete 'this'*
Agreed and implemented.

*L133: "tends to concentrate inland, away from the ice shelf edge" I do not see this spatial distribution in the figure. Can you explain this in more detail? What exactly do you mean by concentrate?*

*L136: would "fast flowing areas' be more precise than "glacier zones"?*
Thank you for pointing that out, we agree and clarified our text with:

**L136:** Areas of fast flowing ice originated from the glaciers Northcliffe, Denman, Scott and Apfel, serve as the clearest examples of areas exhibiting high activeness on Shackleton ice shelf.

L138-139: "Unlike damage values, the activeness parameter does not appear to be directly related to the distribution of accumulated meltwater" This implies that the distribution of meltwater is directly related to damage. This is mentioned briefly at the start of section 3, but is this what is being referred to here? This should be made a little clearer and perhaps this statement softened somewhat, given that the distributions of damage and meltwater accumulation have not been explored in detail and it has not been established that there is a close connection (see my comment on L105-107).
The reviewer is right, this statement is a bit strong. We will produce an additional figure of the distribution of meltwater lakes compared to the observed damage values, to get an indication of their relation.

*L145: In what sense is the ice shelf a prototypical example? This paper provides no comparison to other ice shelves, so if it is typical, this needs to be discussed. And I think prototypical refers to this example in some way being the originator, or the original version of something, which, unless I am missing something, it is not.*
This was not meant to discuss the whole ice shelf: L145 referred to drainages A-E as 'example' where both high damage and high activeness are detected. This might be an effect to English being our second language. We'll change the sentence to "Drainages A to E on the grounding line on West Shackleton are cases where both damage and activeness are high at the lake drainage sites"

*L147: I am not sure what "the glacier tongue" is referring to. Please clarify.*
The Denman Glacier. Implemented

*L157-159: "Although the exact relationship between these metrics requires further investigation across different ice shelves and with more drainage events, it is clear that at least one of these factors, damage or activeness, must be present for lake drainage to occur." As mentioned in the main point above, this statement should be softened. There are examples in Greenland of fracture perpendicular to the background flow direction directly draining lakes on grounded ice. It seems likely the same is possible on ice shelves. A more precise statement restricted to what this dataset tells us about this ice shelf over these three years, given the limitations of the remote sensing datasets and your analysis, is needed here. In other words, not only does the quantification of the exact relationship require further investigation, so does establishing that activeness and damage are a requirement at all.*

These are fair points and we will implement this with care, pending on the added analyses with extended time series.

*L161-163: I suggest deleting this opening paragraph.*

Agreed and implemented.

*L166-167: I suggest deleting the opening sentence of this paragraph.*

Agreed and implemented.

*L171: Delete "Our data reveals a trend:" A trend implies something changing over time. Also, it is unnecessary. I suggest just describing the relationship.*

Agreed and implemented.

*L177: delete "not only confirms but also extends the work of" and put the citation in parentheses. It's not clear to me that this analysis confirms that work. Maybe you can say that it is broadly consistent with it.*

Thank you for your suggestion. We have revised the text to clarify and simplify our findings as follows:

**L177:** Our observations of several drainage events support a model in which tidal forcing plays an important role in modulating supraglacial lake drainage across the ice shelf. This mechanism is broadly consistent with previous findings (Trusel et al., 2022).

*L180: delete "highlight the multifaceted nature of drainage events and"*

Agreed and implemented.

*L182: delete "intricate"*

Agreed and implemented.

*L192-193: "the temporal resolution of satellite passes may not be sufficient to capture all drainage events, especially those of short duration." This is a little repetitive of earlier in the paragraph.*

Thank you for noticing, we will remove this redundant phrase from the paragraph.

*L108-109: "produce different results" can you be more precise with this statement. Changing the thresholds would of course change the results quantitatively, but could it change things qualitatively too?*

Yes, altering the thresholds in the algorithm primarily affects the quantitative outcomes of our study, as some drainage events could then fall into a different 'low', 'medium' or 'high' category, depending on the bounds of these bin classes. However, as long as all ranges of damage and activeness are present on the studied ice shelf, their data distribution and consequently the defined classes are expected to remain relatively similar. Hence it is unlikely to change the qualitative interpretation.

We'll change the sentence to "Consequently, applying different thresholds could yield varying quantitative results, e.g. in terms of classification of damage and activeness."

*L226: this sentence mentions a predictive model for the first time. It isn't clear what this is referring to. What would be the purpose of such a model? I was assuming it would be some kind of parameterization in an ice-sheet model, but the need for real-time data confusing me in that case.*

Thank you for your observation. We acknowledge that the introduction of a predictive model in this context may have been unclear. Our intention was to suggest that combining damage severity, activeness, and meltwater accumulation could serve as indicators for potential drainage sites. In an extreme and ideal scenario, this could be expanded to a model that predicts hydrofracturing before it occurs (hence the near real-time data). Nevertheless, we agree this is probably a bridge to far and will keep it at 'indicator for potential drainage sites'.

L224: When expanding our study to an Antarctic-wide scale, we could assess if the link between drainage events and medium damage, high activeness, and large tidal range occurs on a larger scale. The combination of damage severity, activeness, and meltwater accumulation could serve as indicators for potential drainage sites.

*The discussion: One limitation to the idea of activeness being a control is that fractures could advect into areas that are compressive, but the fractures could remain still perpendicular to flow. This scenario would yield non-zero activeness, but may not be conducive to hydrofracture. This underlines the utility of comparing lake drainage locations to ice shelf stresses and fracture orientation to principal stresses orientations.*

The reviewer makes a very good point. We will include an analysis with calculated strain rates (see earlier comments above) to further analyse our activeness parameter, and will take this case into consideration as well.

*Data availability: It would have been good to have access to the code and data for the review process.*

For the next review process, Google Earth Engine and Python code available through the links below. Please beware those are prelimary versions. For the final submission the repositories will be revisted, cleaned and prepared with documentation.

GEE: https://code.earthengine.google.com/?accept_repo=users/juliussommer/HydrofractureShackleton

Python Github: https://github.com/js-chemE/HydrofractureShackleton_2023

---

## Author Comment (AC3)

**Reviewer 3**

We thank the reviewer for their time and feedback on the manuscript. The valuable comments and questions are carefully considered, and we discuss below how we intent to incorporate their suggestions in the next version of the manuscript.

In short (also considering comments from the other reviewers), we will focus on (a) extending our temporal analysis to increase our sample size of observed lake drainages, (b) extend analysis of our activeness parameter by comparing to the vulnerability map of Lai et al. (2020) and to the strain rates (c) provide more details on methods, specifically the NeRD algorithm and the definition of thresholds and (d) will implement major textual changes to better clarify and align our conclusions to our observations.

For some (minor) edits we have already started implementing changes, and for some comments we can already provide some provisional additional figures in this document. We thank you for understanding this was not feasible yet for all comments.

**General**

The goal of the work is very straightforward and well defined. Basically, there are three main goals: 1) map the overlap of damage and ponds; 2) find the draining ponds and identify in which conditions they occur; and 3) identify triggering events.
It is an interesting study, very important in the study of Antarctica. However, it is hard to define how impactful the findings are, since the methodology lacks more description of the metrics. Furthermore, the analysis should be deeper (see Major Comments).
I think the manuscript has 4 main issues:
1. observations do not support the claims in the manuscript
2. lack of a more robust statistical analysis
3. small number of observations
4. more descriptive methodology and how are the thresholds defined

We understand and acknowledge these concerns. Generally, concern 1-3 stem from the small number of observations and how these were used and analysed. Our study is a conceptual study and we concede that some claims have been too strong. In the next version of the manuscript we'll provide more analysis, clarification and nuance to better align these aspects.

In the next version of the manuscript we will (a) extend the time series to increase the number of observations to support our analysis and claims, to 2015-2024 (adding 6 more years). However, even then the study will still remain relatively small (covering one ice shelf) and the main effort of our revisions will be (b) aimed to rephrase our conclusions to align the observations to the claims. We think, even with the small sample size, this study can add to the scientific discussion of when/where hydrofracturing is actually observed to occur, adding more nuance to a general state of vulnerability of all fractures to hydrofracturing (e.g. Lai et al (2020)). However, we do agree that this study does not provide enough observations to make strong (Antarctic) generalisations, and we will adjust the text accordingly, focusing more on qualitative arguments rather than quantitative.

Considering issue (4): we will put more effort in describing and clarifying the methods. The NeRD algorithm and the thresholds specifically, both for lake detection and damage detection, will be better explained, and a supplementary section will be included to address this. Our code will be made available:

For the next review process, Google Earth Engine and Python code available through the links below. Please beware those are prelimary versions. For the final submission the repositories will be revisted, cleaned and prepared with documentation.
GEE: https://code.earthengine.google.com/?accept_repo=users/juliussommer/HydrofractureShackleton
Python Github: https://github.com/js-chemE/HydrofractureShackleton_2023

**Major Comments**

**2.1 Sample size**

The authors used a time span of 3 years for the analysis performed. They found 13 events, which is a small sample size. Why do you not cover a larger time span to have more data? In addition, covering other regions would be beneficial. Furthermore, the definition of threshold values seems quite arbitrary, indicating that they fit only to these present specific conditions.

Also, the authors highlight the need for more sophisticated statistical approaches. This is a major concern for me, because not only the dataset is small, but it also lacks meaningful statistical analysis.

We agree that the sample size is small. For the next version of the manuscript, we'll extent the time span of the study to cover 2015-2024 (governed by the availability of Sentinel-1 for consistent damage mapping). While this adds more datapoints, we don't expect an order of magnitude change in available observations of rapid lake drainages, and therefore we will also focus the text on a more qualitative analysis. The analysis will be extended by using a more comprehensive comparison to the vulnerability assessment of Lai et al. (2020), for which some preliminary data is included in this document below.
Considering the thresholds: we presume the reviewer is talking about the lake drainage threshold (80% of volume). This threshold has been chosen consistent with previous literature (see reviewer #1 for detailed discussion and references), and is therefore not chosen specifically for our conditions. Nonetheless, we will expand on this and include an uncertainty estimate to the number of drainages detected when varying to 75-90% of volume (similar as Miles et al., 2017).

[Figure]

**2.2 Not supportive data**
I think the data presented do not support the conclusions drawn. In fact, the authors repeatedly make an affirmation and draw it back after a few sentences, for example:
L139-140 = "However, all of the detected lake drainage events occur in areas of the ice shelf classified as medium to highly active"
And L151-L153 = "For example, drainages H, K, and M took place in areas with relatively low damage but high activeness, while drainages A, F, and G occurred in the least active regions, yet showed high levels of damage."
Furthermore, analysis of Figure B1 reveals that 1 of the events is in a low activeness area, undermining the claim in L139-140.

We concede that the wording of these claims are too strong, and inconsistent. To clarify, indeed the first sentence (L139-140) should not be 'all drainage events', but '10 out of 13'. Our sincere apologies for this error. We will adjust accordingly, and by extending the timeseries to 2015-2024, these descriptions of observations and the conclusions drawn will undergo major revisions.

**2.2.1 Activeness parameter**
The authors conclude that "all of the detected lake drainage events occur in areas of the ice shelf classified as medium to highly active", but they sum up to 90% of the studied region. Taking into account that they have only 13 drainage events, their conclusion is not supported by the data since further analysis should be made.

Even if drainage events are correlated to activeness, it could be the case that activeness is actually related to damage (meaning that damage is higher when it is caused by the flow, which is very plausible) and that drainage events are mainly driven by damage (also very plausible).

We completely understand the critical standpoint against the low amount of samples, and, again, will alleviate these concerns by extending the timeseries with 6 more years. The conclusions will be adjusted with those results, and more care will be taken to align the weight of the conclusions to the observations.

A note about the damage signal strength: this is uncorrelated to the ice flow, as it is purely based on how clear the damage feature is visible in the image, as a stark white line in a dark field on radar images (in optical images it is a dark line on white), which is mainly dependent on the look angle of the Sentinel-1 sensor and the amount of noise/speckle, where backscatter of the radar is generally highest for steep vertical walls. The strongest damage signals are actually found near and at the ice front, where large full ice-penetrating rifts yield the highest contrast between the ice and the ocean.
Apart from this, the activeness parameter is indeed correlated to damage. It is based on the orientation of the detected damage, not on the strength of the detected signal, but still, there has to be a detected damage feature to get an activeness assessment. The intent of the activeness parameter is to shift the focus of the detected damage signal strength: you can find small yet opening crevasses ('active') as well as large but stationary rifts ('passive').

*2.2.2 Section "Lake Drainage Events in Periods of Increasing Tidal Heights"*
As reviewer #1 said "6/11 drainage events in 2019 (more than half) started during the lowest (or even descending) phase of the tidal cycle (drainages M, L, F, H, J, E)". This simply invalidates the sentence: "Our findings unveil a compelling narrative of ice shelf dynamics, revealing an intricate interplay between tidal forces and supraglacial lake drainage events".
Furthermore, in a hypothetical case where all the drainage events occur in an ascending amplitude phase of the tidal, it will not imply what is said: if you have thousands of lakes in a ice shelf, and each time only a small fraction of lakes drain, the likelihood of a drainage event triggered by tidal flexure would be higher in the crest of the amplitude phase.

We refer to our response to reviewer #1, which we will repeat here as well: we would like to clarify that indeed drainage events M, E and K 'start' in the descending phase (3/13) and 4/13 in the lowest phase. However, with this, it is important to realise that the drainage events shown in figure 3 occur *between* the detected dates t1 and t2, since we only detect 'lake is present' at t1 and 'lake has drained' at t2 – it is not a draining that starts at t1 and ends at t3. Therefore, Figure 3 does (in our opinion) clearly suggest that drainage does not occur in the descending phase. We do acknowledge that we cannot differentiate if the drainage occurs in the lowest part of the cycle or the ascending part, so will adjust the text accordingly. As for the higher likelihood of a drainage event triggered in the crest of the amplitude phase: this is not reflected in any of our observed drainages.

**Minor comments**

**3.1 Methodology clarification**
Some clarifications are needed on the two main metrics used in the study. First, I agree with reviewer #1 about the lack of information on how damage is calculated. Usually it is inverted from
$$\mu = (1 - D)B2\dot{\varepsilon}^{n-1}ne \quad (1)$$
where $\mu$ is the ice viscosity, $D$ is damage (which you want to invert), $B$ is the ice rigidity, $\varepsilon_e$ is the effective strain rate, and $n$ is the flow law exponent. I think it is needed to specify how damage is calculated from remote sensing. Also, if possible, it would be interesting to relate the damage calculated in the present work (which reviewer #1 suggests changing the name to "satellite-derived damage" and I agree), and damage calculated from Equation 1.

We see that in lieu of brevity the methods have been too concise, and will add more details on how exactly damage is calculated. In short, the NeRD method consists of the following steps: (i) create cut-out windows from the image (for which we use 10x10 pixels), (ii) apply the Normalised Radon transform to these

windows, (iii) extract dominant feature signal strength and orientation for every window, (iv) quantify the damage signal value by removing noise from the signal and (v) postprocessing. In the post-processing step we clipped the product to the ice shelf bounds.

As the NeRD method is a published algorithm (Izeboud and Lhermitte, 2023), the description will remain short and to the point, as for extended sensitivity studies and evaluation of the method we refer to that publication. Nevertheless, we will include the produced annual damage maps (also included below, before downsampling to 3 km) and an assessment of the changes in detected damaged area for every year in the supplementary material.

We do find calling it satellite-derived damage a good suggestion and will implement this term as well as referring more strictly to 'damage signal' or 'detected damage' in the manuscript.

Lastly, we agree that it would definitely be interesting to relate the calculated damage maps to damage calculated from damage mechanics models, and to our knowledge this is an active field of research – refer to e.g. Gerli et al. (2024) and De Rydt et al. (2021) – but we consider it out of scope for this study.

- *Izeboud, M. and Lhermitte, S.: Damage Detection on Antarctic Ice Shelves Using the Normalised Radon Transform, Remote Sensing of Environment, 284, 113 359, https://doi.org/10.1016/j.rse.2022.113359, 2023.*
- *Gerli, C., S. Rosier, G. H. Gudmundsson, and S. Sun. 2024. 'Weak Relationship between Remotely Detected Crevasses and Inferred Ice Rheological Parameters on Antarctic Ice Shelves'. The Cryosphere 18 (6): 2677–89. https://doi.org/10.5194/tc-18-2677-2024*
- *De Rydt, J., R. Reese, F.S. Paolo, and G. H. Gudmundsson. 2021. 'Drivers of Pine Island Glacier Speed-up between 1996 and 2016'. Cryosphere 15 (1): 113–32. https://doi.org/10.5194/tc-15-113-2021*

[Figure]

*Detected damage by the NeRD method at 300 m resolution, before downsampling to 3 km*

**3.2 Activeness vs. damage**

*I think a further analysis of the relationship between damage and activeness is required. It can be the case that the relationship between them is high, so any relationship between activeness and drainage occurrences is only due to damage.*

The activeness parameter is correlated to damage, but only in the sense that it is determined in areas where damage is detected (binary): it is based on the orientation of the detected damage, not on the strength of the detected signal. It shifts focus to damage features that are likely undergoing change (opening, widening) from features that are stationary/passive. It is an interesting point though, that there could be areas of 'activeness' without visible damage features to show for it. To address this we will add a comparison of the

strain rates, damage, and activeness parameter, as we expect this will shed more light on where and when damage with high activeness is found.

**3.3 Motivation**

*The first sentence of the article ("Surface lake drainage can destabilize ice shelves, occurring either slowly via supraglacial channels or rapidly through crevasses") makes me wonder if you are not studying the opposite. Instead of analyzing how damage influences lake drainage, should not you analyze how lake drainage influences damage? Otherwise, if you really want to analyze how damage influences lake drainage, you should motivate that in the introduction. I think you are putting the cart before the horse.*

Thank you for this insight, it's important to us that the introduction is very clear. We are mainly analysing the place and timing of lake drainages, and in that sense we are analysing how damage influences lake drainages: we hypothesised that just 'having' damage features does not necessarily lead to hydrofracturing – since damage is so abundant on many ice shelves in antarctica, and hydrofracturing is less widespread.

**We will edit the introduction to (subject to small editorial changes):**
Surface lake drainage can destabilize ice shelves, occurring either slowly via supraglacial channels or rapidly through crevasses driven by the weight of the water – a process known as hydrofracture (Nye and Perutz, 1957). While Greenland's lake drainages are relatively well-studied (Williamson et al., 2018a; McMillan et al., 2007), much less is known about similar processes in Antarctica. Understanding when and how surface lakes drain is crucial for assessing ice shelf stability. Previous studies, such as Trusel et al. (2022) have shown that lake drainages on the Amery Ice Shelf are linked to high-amplitude tidal cycles. Moreover, Lai et al. (2020) have shown large areas of Antarctica's ice shelves that are vulnerable to hydrofracturing if (existing) crevasses are inundated with meltwater. However, given the widespread presence of crevasses and other damage features on Antarctic ice shelves, it remains unclear to what extent pre-existing damage influences the likelihood and timing of lake drainage events. Here, we use observations of lake drainage events from remote sensing data to study their place and timing, examining whether damage alone is sufficient to indicate a potential of hydrofracturing, or if additional conditions, such as tidal forcing, are necessary to initiate lake drainage

**Specific comments**

*L16: Is it the first time that "activeness" is used? If so, say that the manuscript introduce this concept. Otherwise, make a reference.*
Thank you for your comment. The concept of activeness is newly introduced by us. And we will adjust the sentence accordingly:

**L15:** We therefore hypothesize that, apart from using the presence of damage features, another metric is needed to indicate a likelihood for occuring lake drainages. Specifically, we propose that in addition to the presence of damage (open crevasses, fractures, and rifts), a measure of the activeness of the damage feature (i.e. crevasse opening or propagation) can be used to identify where lake drainages are likely to occur on an ice shelf.

*L44: Do you advect the features when you merge the images in the mosaic?*
Thank you for your question. We did not account for advection when creating the mosaics. Since each mosaic covers a maximum of 10 days, the only area where advection would be significant is at the fast flowing Denman Glacier (max speeds of ~1500 m/yr (Miles et al. 2020) in its center, ~4.6 m/day, just enough to have 1 pixel of advection in the 30 m Sentinel-2 images in the selected period) -- ice flow speed quickly drops to <= 500 m/year to the sides of the ice tongue, where the majority of the melt lakes are observed. Moreover, in this short period of time the amount of repeat satellite overpasses is very limited, and we get just the minimum amount of image to stitch them together for a domain-covering mosaic, thus having almost no impact of advection on the mosaic calculation. We'll include a supplementary figure in the next manuscript to show the overlap of the used satellite images to construct each mosaic, and clarify in the text that advection is not taken into account.

*L50: This sampling frequency (once a year) is very different from the optical image that you are going to contrast later on. How do you deal with that?*

The damage maps are derived outside the melt season to ensure reliable detection. They show minimal year-to-year variability (as it can be seen above), and we will include them in the supplementary material.

*L52: It is not clear where you use the velocity field. For sure for the "activeness" calculation, but do you also use to transport the features? Make it clear near the description of the velocity field calculation.*

Thank you. Agreed and we will adjust the sentence as follows:

**L52:** Ice flow velocity data for 2019 are sourced from the ITS_LIVE campaign (Gardner et al., 2020) and used for the calculation of the activeness metric.

*L61: Regarding the threshold 1800 m2 you make a citation for this value, but what is the reasoning of using this threshold?*

Thank you for the comment. All subsequent steps where executed as indicated by the reference. But in order to clarify the purpose of reach step we have split the sentences:

**L61:** Outliers are removed in a subsequent step if they are not located on ice mass or have a misinterpreted depth of less than 0 m (Williamson et al., 2018). To minimize further noise, lakes with a surface area of less than 1800 m2 (2 or 18 pixels of L8 and S2 imagery, respectively) are removed as suggested by Williamson et al. (2018).

*L65: Why 80%? I can also imagine 50% as a massive drainage event. this looks like a random choice, that you need to pick one, but if you lower the threshold, you would have many more drainage events, increasing the data you can use to infer, since 13 drainage events are not many.*

This is in line with other literature, following Doyle et al.(2013), Fitzpatrick et al. (2014), Miles et al. (2017), Williamson et al. (2017). These references will be specified in the text.

- *Doyle, S. H., Hubbard, A. L., Dow, C. F., Jones, G. A., Fitzpatrick, A., Gusmeroli, A., Kulessa, B., Lindback, K., Pettersson, R., and Box, J. E.: Ice tectonic deformation during the rapid in situ drainage of a supraglacial lake on the Greenland Ice Sheet, The Cryosphere, 7, 129–140, https://doi.org/10.5194/tc-7-129-2013, 2013*
- *Fitzpatrick, A. A. W., Hubbard, A. L., Box, J. E., Quincey, D. J., van As, D., Mikkelsen, A. P. B., Doyle, S. H., Dow, C. F., Hasholt, B., and Jones, G. A.: A decade (2002–2012) of supraglacial lake volume estimates across Russell Glacier, West Greenland, The Cryosphere, 8, 107–121, https://doi.org/10.5194/tc-8-107-2014, 2014*
- *Miles, K. E., Willis, I. C., Benedek, C. L., Williamson, A. G., and Tedesco, M.: Toward monitoring surface and subsurface lakes on the Greenland Ice Sheet using Sentinel-1 SAR and Landsat 8 OLI imagery, Front. Earth Sci., 5, 1–17, https://doi.org/10.3389/feart.2017.00058, 2017*
- *Williamson, A. G., Arnold, N. S., Banwell, A. F., and Willis, I. C. (2017). A Fully Automated Supraglacial lake area and volume Tracking ("FAST") algorithm: development and application using MODIS imagery of West Greenland. Remote Sens. Environ. 196, 113–133. doi: 10.1016/j.rse.2017.04.032*

*L67: Was not the are threshold 1800? Furthermore, why do you use these threshold? Give a reason and use the citation. Only the citation is not enough.*

Thank you for your comment. We use two sets of thresholds in our analysis. The 1800 m² threshold is applied to minimize noise by excluding very small features that are likely spurious. In contrast, the 54,000 m² threshold is used to focus on lakes that are large enough to potentially drain and impact the ice shelf. This 54,000 m² value corresponds to about 60 pixels in Landsat 8 imagery, a size considered significant for water volume and hydrological impact (Williamson et al. (2018a)).

*Williamson, A. G.; Banwell, A. F.; Willis, I. C.; Arnold, N. S. Dual-Satellite (Sentinel-2 and Landsat 8) Remote Sensing of Supraglacial Lakes in Greenland. The Cryosphere 2018, 12 (9), 3045–3065. https://doi.org/10.5194/tc-12-3045-2018 a.*

*L72: you excluded 20 out of 25, so you have 5 drainage events. How then you have 13 drainage events in your results?*

Apologies, but we are unsure where the count of 20 excluded events comes from. The manuscript states in **L72:** *"Twelve out of twenty-five events are removed, as they are judged to be refreezing lakes rather than draining lakes."*

This results in 13 events.

*L78: Add "resolution" after "300 m".*

Agreed and implemented.

*L83: Missing citation. Makes sense to compare the angle of the fracture to its orientation, but quantifying it can be tricky. Is there any supporting studies for the use of that values?*

We apologise for any confusion: the angle of the damage feature is a result of the NeRD method (specified just below this sentence). We will clarify in this sentence ("The obtained damage orientation from the NeRD algorithm (Izeboud and Lhermitte, 2023) is used to identify areas with a likelihood of active damage development, ..."). Moreover, as we will expand the explanation of the NeRD algorithm to provide more clarity on how damage (and its orientation) is detected in the method section beforehand, this will further aid clarity.

*L88: If activeness is binary, how you produce an image like Figure 2 b)? It do not seems like a product of downsampling.*

Thank you for your comment. Our activeness metric is indeed binary (0 or 1) at the original 300 m resolution. However, when we downsample by a factor of 10 using an average resampling method, each 3000 m pixel then represents the average (or proportion) of active pixels in that block, which naturally yields continuous values between 0 and 1. This will also be clarified in the text.

*L88: I don't see the reason of donwsampling it.*

It can be very tricky to properly assign which damage features should be linke to which lake drainage events based from these remote sensing observations, and what the appropriate lengthscale of such influence is. Furthermore, we hypothesize that an area of active damage might be indicative of a general structurally weakened ice zone, which might facilitate lake drainages through previously undetected (small) fractures. To capture this, we translated the detected damage and activeness parameter into a less localized representation, reflecting the overall integrity of the ice over a larger area.

*L89: Why normalizing? You can say directly that damage varies between 0 and 1 and everything is already normalized.*

Thank you for the question. We are using the NeRD algorithm (Izeboud and Lhermitte, 2023) which outputs damage values between 0 and 0.5. We then normalize these values using the maximum derived from the Shackleton Ice Shelf, which standardizes the data into a 0 to 1 range for easier comparison across the study area.

*L89: Add "resolution" after "3000 m".*

Agreed and implemented.

*L101: Add "total" in "with total maxima".*
Agreed and implemented.

*L116: I think the definition of the thresholds should go to methods with a further explanation of the threshold used.*
Agreed, we concur that for the sake of brevity we left out too much, and we will implement this.

*L119-124: This is a very sounding result, supporting the hypothesis of a strong relationship between drainage events and damage. However, I would expect the same analysis regarding the activeness. The distribution of 10%, 71%, and 19% does not allow this analysis. Contrasting the area distribution and Figure B1, and do not see a strong relationship between activeness and drainage events. I would say that most of the signal of drainage events are due to damage.*
This is insightful, and we will take this into consideration when we revise the analyses with the extended time series, which may or may not change these results.

*L129: Same comment as for damage.*
Understood, the thresholds will be more extensively described.

*L139: This conclusion is not surpirse. It sums 90% of the studied area.*
The reviewer is correct. We'll revise accordingly with the extended timeseries.

*L151: What do you mean by "parallel trend". Be more specific. **L151-153:** This goes against the phrase: "However, all of the detected lake drainage events occur in areas of the ice shelf classified as medium to highly active". **L156:** You previously said the opposite. **L158:** As far as I understood, it is impossible to drainage event to occur without damage. Be more precise with this statement.*
Thank you for your comment. We agree that the wording needs to be more precise and consistent regarding the derived categorizations of damage and activeness. We have revised the text as follows:

**L151 – L159:** Intriguingly, activeness and damage do not always follow the same behaviour, suggesting that these two factors influence lake drainage in a more complex manner. For instance, drainages H, K, and M occurred in areas with ==medium== damage but ==high== activeness, whereas drainages A, F, and G were found in regions with ==medium== activeness yet exhibited ==high== levels of damage. By integrating the dominant orientation of fractures relative to ice flow into our activeness metric, we capture an independent characteristic of ice shelf damage that can vary from the overall damage intensity.

*L161: I agree with reviewer #1 regarding the drainage events with respect to the ascending phase of tidal cycles.*
We understand the concern. We would like to clarify that indeed drainage events M, E and K 'start' in the descending phase (3/13) and 4/13 in the lowest phase. However, with this, it is important to realise that the drainage events shown in figure 3 occur between the detected dates t1 and t2, since we only detect 'lake is present' at t1 and 'lake has drained' at t2 – it is not a draining that starts at t1 and ends at t3. Therefore, Figure 3 does (in our opinion) clearly suggest that drainage does not occur in the descending phase (even though, indeed, crevasse might be more prone to opening in that phase!). We do acknowledge that we cannot differentiate if the drainage occurs in the lowest part of the cycle or the ascending part, so will adjust the text accordingly.

*L187: I think you do not have drainage events that last hours. If this is the case, remove the "few hours".*
You are right and we'll remove this.

*L190: Here you say that it is difficult to assign the drainage events to hydrofracturing, but in the discussion you did this.*

Thank you for your observation. We acknowledge that our previous wording may have been unclear. While we can identify the occurence of drainage events (i.e. detecting a drained lake) using satellite imagery, we don't know the exact timing and speed of the drainage that occured between the satellite overpassess. This is the distinction we were trying to convey, and we'll edit the text to clarify this.

*L197: If NeRD can not identify individual fractures, how then you measure the orientation of the fractures to calculate the activeness?*

Apologies, this wording is misleading. NeRD returns one value for damage signal strength and one for damage orientation for every processing window of 10x10 pixels (30 m per pixel). It does not, however, return the exact location of the detected feature within the window, and neither its width or length. It is also possible there are multiple crevasses within the window, for which case the algorithm favors the feature with the strongest contrast (see Figure from Izeboud and Lhermitte (2023) below). So, what we mean is actually that NeRD does not detect the exact outlines of individual features.

[Figure]

**Fig. 2.** Idealised scenario's to illustrate the differences between the Radon transform with and without normalisation. Panel a–e represent different scenario's to which the Radon transform is applied: a1–e1 show an idealised window with a hypothetical crevasse, a2–e2 show the corresponding 2-D feature space $R(\rho, \theta)$ without and with normalisation (respectively top and bottom), and a3–e3 the signal response $\sigma(\theta)$ with and without normalisation — from which $\sigma_{crev}$ is extracted (black dot).

*214-216: This sentence is another major concern for this study.*
We understand the concern. Again, this comes back to the low sample size (this sentence refers to the 2 of 13 drainages that occur in low activeness areas), and we expect to improve upon these assessments with additional observations by extending the timeseries.

*I would appreciate a hexabin graph with activeness and damage in the axes. This would allow us to see the correlation between both metrics and the occurrence of drainage events.*

Thank you for the suggestion, we brought two plots from our previous discussions.
We could add something similar to Figure 2 and address the distribution of detected drainage events vs the distribution damage and activeness.

[Figure]

Figure is showing detected activeness versus logarithmic damage. The dots are the detected drainage events, dashed lines are the respective limits for medium and high characteristics. The secondary axis show the cumulative ice shelf coverage of damage and activeness of the melt season 2019/2020.

[Figure]

Figure is showing detected activeness versus logarithmic damage (please note that for swift plotting we applied the logarithm instead of making the scale logarithmic). The dots are the detected drainage events. The dashed lines are the respective limits for medium and high characteristics. The histograms show the distribution over the ice shelf damage and activeness of the melt season 2019/2020.

*Table A1: Bring it to the main body of the text, it is too important. I suggest including two more columns: Classification of damage and activeness (low, medium, high).*

Good idea, we'll add the extra columns. We agree that it would be good to have this in the main body. However, the Brief Communications format only allows for three display items (tables/figures) so we are very limited in our flexibility here, unfortunately.

*Figure 1: Define LIMA as an optical imagery mosaic from Landsat.*

Agreed and implemented.

*Figure 2: Add ", respectively" at the end of the first sentence.*
Agreed and implemented.

---

## Author Response (AR1)

**Reviewer 1**

We thank the reviewer for their time and feedback on the manuscript. Their valuable comments and questions have been carefully considered, and we discuss below how we have incorporated their suggestions in the new version of the manuscript. The manuscript has undergone major revisions, which we will introduce briefly here, before providing more detailed response to the reviewer and their concerns:

- We have extended our temporal analysis to increase our sample size of observed lake drainages, from 2015-2024, adding 6 more years. We now observe 21 lake drainage events (instead of 13).
   There have been no major changes to the results by including these new events, however, we have made major textual edits to provide more nuance to the previously too strongly made claims.
- We have incorporated a comparison to the vulnerability map of Lai et al. (2020), providing insights into where they estimated the ice shelf to be vulnerable to hydrofracturing and where we observe lake drainage events. We observe quite a number of drainage events in 'non-vulnerable' areas, that are characterized mostly with relatively moderate activeness and high damage, compared to the vulnerable areas having drainage events of high activeness and high damage.
- Manuscript Figure 1 now includes examples of detected drainage events.
- Manuscript Figure 2 now includes a panel with the comparison to Lai et al.'s detected vulnerability as well as plots of data distribution that were previously part of supplementary material.
- Manuscript Figure 3 is only adjusted to incorporate the new observations.
- We have also made textual changes to clarify the methods.

**General Comments:**

An important point to clarify is the definition of damage. Initially, damage is defined in modeling as a variable of the enhancement factor, which modulates the ice's fluidity. This modeling variable is defined between 0 and 1 and can be adjusted to best fit the observed ice flow. Here, the authors completely recalculate this damage variable from satellite observations, independently of a model or flow velocities. It is not entirely clear how these products could be directly used in a flow model. Some studies have suggested that the crevasses derived by NerD do not match the damage modeled in an ice flow model (Gerli et al., 2024). Therefore, to avoid confusion between the terminology used by modelers and that of this paper, I suggest using the term "satellitederived damage" or something similar throughout the manuscript.

Indeed, as the reviewer states, our use of 'damage' is different from the definition in damage mechanics models. It is also not a parameterization that can directly be used in flow models since there is no quantified translation from damage signal to fracture depth (as is explained in Izeboud and Lhermitte (2023)). We tried to more explicitly define 'damage' in the manuscript as 'visible features of damage at the ice shelf surface' as an umbrella for crevasses, rifts and fractures.

We appreciate the suggested term, and have adjusted the manuscript to use 'satellite-derived damage' or "detected damage" when discussing our observations:

- **L21-22** "Specifically, we propose that in addition to the presence of visible damage (open crevasses, fractures, and rifts), a measure of the `activeness' of the damage feature (i.e. crevasse opening or propagation) can be used..."
- L42 "These drainage events are compared to satellite-derived damage and..."
- Small edits throughout manuscript

Regarding the calculation of damage, why didn't the authors use the same optical images as for lake detection? It seems that NerD also works with this type of image. The paper lacks details on the time coverage of Sentinel-1 vs. Sentinel-2 images. Do the dates align perfectly? What is the time delay? There are also unclear areas regarding damage calculation: only one map is presented—is it a mosaic? Over what period was it calculated? Why don't the authors present a time series of damage alongside the time series of lake drainage? This would better specify the exact timing of events, particularly concerning tidal cycles: for instance, does damage increase with tides? Does this increase precede drainage events?

While NeRD is compatible with both optical and radar imagery, we opted for Sentinel-1 radar data for damage detection. This choice was driven by the fact that optical images, including those from Sentinel-2, are affected by cloud cover, limiting their usability for consistent monitoring. In contrast, Sentinel-1 provides reliable coverage over short time periods, ensuring a more complete dataset for our analysis.

Regarding the damage map, we acknowledge that our methods section could have been more explicit. We created a single domain-covering map by mosaicking individual Sentinel-1 images from November 1-10 of each year that cover the Shackleton Ice Shelf, which was then processed with NeRD.

The time period for damage assessment does not specifically overlap with the lake drainage events. This is because NeRD detects linear features based on grayscale contrasts, and is therefore sensitive to edges between water and snow. To minimize potential interference from surface meltwater, we used images from before the melt season, which also reinforces the use of Sentinel-1 over Sentinel-2 for this purpose.

In terms of damage characterization, NeRD is designed to detect large-scale damage features (>100 m) and assess the spatial extent and growth of damaged areas. However, it is not suited for tracking fine-scale crevasse development or deepening. The detected damage patterns by NeRD are quite consistent throughout the years. Annual maps have been included in the supplementary figures, see an excerpt below.

Figure R1 Example of annual damage maps, detected by the NeRD method at 300 m resolution, before downsampling to 3 km

A second point concerns the features detectable via the NerD algorithm. Looking at Figure 1A, it seems the algorithm effectively detects dislocation zones with wide surface crevasses, as observed near Denman's grounding line or north of Denman Ice Shelf. However, examination of a Landsat image (see below) reveals that almost the entire Denman Ice Shelf is heavily fractured by basal crevasses, unlike the Shackleton Ice Shelf. Yet, damage maps show very similar low damage values, suggesting an underestimation of basal fracturing. Given this limitation, I think the interpretation of results should be revisited, and this should be mentioned in the discussion, specifically in the sense that ice shelf can still be damaged from below, if it is not detected by your method. Additionally, a significant portion of the Denman Ice Shelf's front appears to be detached from Shackleton, more so than shown in Figures 1 and 2. Thus, some of the damage signal may no longer belong to the ice shelf and could merely represent calving icebergs. Analyzing lake evolution in this sector is therefore limited. The Shackleton Ice Shelf's frontal boundaries are also ambiguous (see image below), with a border region between sea ice and mixed ice that could bias the damage analysis. How are you dealing with these regions?

We acknowledge the limitation that NeRD tends to underestimate basal fracturing, as basal crevasses are often represented as low damage values or may not be detected at all. We have clarified this in the discussion, L208: "By using satellite-derived damage, these analyses also exclude the influence of basal fractures."

Regarding the Denman Ice Shelf front, we appreciate this consideration and agree that it can be ambiguous what is still ice shelf or ice berg. The damage maps were clipped to ice shelf fronts that were manually delineated based on Sentinel-1 observations of each respective year. However, the impact of any inaccuracies in the ice shelf front positioning is limited due to the following factors:

- (a) The velocity maps (from MEaSURES ITS\_LIVE campaign) were masked more strictly than the damage maps, effectively removing the outermost portion of the Denman ice front, as seen in Figure 2 in the manuscript.
- (b) Only one drainage event (event F) was identified in that region, and all detected drainage events were manually verified to exclude false positives.

L200-215: How does the coarser resolution affect your interpretation? Drainage is a highly localized phenomenon. When downsampling, you might "leak" damage values located far from drainage events, especially with a factor of 10. Why was a factor of 10 used? Were smaller downsampling values tested unsuccessfully? If so, this could indicate that the correlation between damage and lake drainage events is not as strong as suggested, and tuning the data by downsampling may not be the correct approach? Furthermore, most figures shown in this paper are very general and synthetic, for the sake of the "brief communication" format. This significantly limits the analysis of results. For example, it is important to include a zoomed-in or more detailed figure focusing on a lake drainage event that clearly shows the relationship with damage/crevassed regions (or a new panel of Fig 2). Similarly, it would be important to include a Sentinel-1 image with the retrieved damage (at least in an appendix) and the visually observed fracturing. I am unsure of the best solution: either move the paper to the regular format of The Cryosphere or add one or two figures to the appendix.

There's multiple aspects we considered. Firstly, NeRD does not delineate individual crevasses, but returns the presense of a crevasse in a 300-m area, so there is already an inherent 'leakage' when matching drainage events to detected damage. Secondly, we indeed first tested without further downsampling approaches and found and found that correlating lake drainages with detected damage was very difficult, mainly due to the abundance of detected damage and the very limited observations of actual drainage events. With some visual inspection we noticed that often damage features next or close to the lakes were detected, but not specifically on the location of the lake. Since NeRD detects features of >100m, we think it is likely that the drainage events occur through smaller undetected fractures. We therefore downsampled the data to get a sense of the 'surrounding area' on drainage events.

This has been clarified in the text, L105-108: "From these observational products we cannot prove causality between individual damage features and specific drainage events. Furthermore, it's possible for the drainage to occurs through a fracture that is not visible in the 300 m maps. We therefore use the damage and activeness maps as an indication of a general structurally weakened ice zone, which we hypothesize to favor lake drainages through undetected or new (small) fractures. For this reason, we downsampled the damage and activeness maps to inspect the overall integrity of the ice for a larger area surrounding drainage events. Both 300 m maps are downsampled ... "

We have added a zoomed-in figure of two lake drainage events to Figure 1 in the manuscript (see Figure R2 in this document).

Figure R2 Two examples of detected lake drainage events. The drainage events are the lakes at the center of the image.

Concerning the tidal analysis, the authors argue that drainage events always occur during the ascending phase of tides. Looking at Figure 3, I would argue this is a bit of a stretch. Indeed, 6/11 drainage events in 2019 (more than half) started during the lowest (or even descending) phase of the tidal cycle (drainages M, L, F, H, J, E). The same applies to 2020, which saw only one drainage event. This does not undermine the paper's conclusions, as it remains plausible that changes in stress on the ice shelf with tidal cycle, and thus damage, favour these drainage events. For example, could the authors argue that the descending phase of the tidal cycle might be even more prone to crevasse opening due to flexure, whereas the ascending phase might favor crevasse closure?

We understand the concern. We would like to clarify that while indeed drainage events M, E and K 'start' in the descending phase it is important to realise that the drainage events shown in figure 3 occur between the detected dates t1 and t2, since we only detect 'lake is present' at t1 and 'lake has drained' at t2 – it is not a draining that starts at t1 and ends at t3. Therefore, Figure 3 does (in our opinion) suggest that drainage does not seem to occur in the descending phase. This is further corroborated with the new drainage events.

We do acknowledge that we cannot differentiate if the drainage occurs in the lowest part of the cycle or the ascending part, and that we cannot pinpoint the exact moment of drainage, so have adjusted the text accordingly:

**L184-186**: "We compare the drainage time-windows to tidal data (Figure 3), and indeed find a clear pattern: the majority of drainage episodes aligns with the ascending phase of tide cycles. Although we cannot determine the exact drainage date, only a snapshot before and after the event, few drainages seem to have occurred on the descending phase"

As for the flexure: yes the ascending phase favors closure of surface crevasses, nevertheless the drainages are for the majority observed in this phase. It might mean that the influence of basal crevasses is high (which would be prone to open during ascending phase).

This has been included in the manuscript discussion, **L221-224**: "Our results show that the observed lake drainages coincide mostly with the ascending phase of tidal amplitude. This phase would correspond with closing of surface crevasses and opening of basal crevasses, indicating that the trigger for drainage events might not be depending on the stability of surface crevasses only."

**Specific Comments:**

L46: Does lake advection with ice flow affect your mosaic calculation in any way?

No, this effect is negligible. The mosaics are calculated on a very short period of time (8-10 days) and thus the only area where advection would be significant is at the fast flowing Denman Glacier (max speeds of ~1500 m/yr (Miles et al. 2020), ~4.6 m/day, just enough to have 1 pixel of advection in the 30 m Sentinel-2 images in the selected period) -- ice flow speed quickly drops to

L127: If the activeness value mainly reflects shearing, why not directly use shear strain rates? What is the added value of activeness?

Figure R3 Distribution of detected damage and activeness parameter, now included in Figure 2 in manuscript

While shear strain rates provide valuable information about the stress environment within an ice shelf, they do not capture the behaviour of observed damage features. Damage development (crevasse or fracture opening) is strongly linked to strain rates, but they advect and rotate with the ice as well. As a result, observed damage features might be found in areas downstream of where they formed, and might have become 'passive'. The activeness parameter is aimed to account for this, using the observed orientation (angle with respect to flow) to infer if the detected features are in a position that favours crevasse opening.

We clarification in the text where the activeness parameter is introduced:

**L97-100:** "In addition to detecting the presence of surface damage features, we infer if the features are likely actively developing and opening (`activeness') or if they are passive (apart from advecting with the ice flow). The obtained damage orientation is used to identify the potential activeness by comparing the damage orientation to local ice flow angle, to infer if the feature is in a position that favors crevasse opening."

**L137: What do you mean by "more pronounced"?**

We were referring to regions within glacier zones where ice flow velocities are particularly high. This has been reworded, the sentence now reads, **L157**: "High activeness is found most clearly where fast flowing ice experiences shear stress from adjacent slower-moving or stationary ice masses, leading to mix-mode crevasse opening (Colgan et al., 2016). Areas close to the Northcliffe, Denman, Scott and Apfel glaciers serve as the clearest examples of areas exhibiting high activeness on Shackleton ice shelf:"

**L150-155: Can you clarify how you can have low damage but high activeness?**

In essence, high activeness indicates the potential for further damage development (crevasse opening or deepening) due to the local stress environment, while low damage simply reflects the current state. You can find small yet opening crevasses ('active') as well as large but stationary rifts ('passive'). This distinction highlights that activeness is more about the dynamics and stress patterns in the ice shelf rather than the extent of existing damage.

L195: Do we really need more data and satellites? With Sentinel-2 providing data every 5 days, Sentinel-1 every 6–12 days, and Landsat-8-9 every 15 days, what is your temporal sampling of drainage events when combining these data? What is the finest temporal scale you could achieve?

The reviewer makes a good point, as more is not always better, and we might instead focus on a smarter use of the data that is available. In any case, hydrofracturing or rapid lake drainages occur on very short time scales (some even under 24 h) and being able to sample within a short timeframe would be beneficial. In our case, we utilized mosaics with a period of 8 to 10 days, which represents the finest possible temporal resolution we could achieve due to the relatively high cloud coverage during the Antarctic summer. For other regions with less cloud cover a finer temporal resolution can probably already be achieved, or by integrating radar (Sentinel-1) in the detection approach: e.g. in West Greenland, Miles et al. (2017) achieved a 3 day interval by combining Sentinel-1 and Landsat-8.

We've revised the text: "To accurately identify and attribute rapid drainage events to hydrofracture mechanisms, a shorter observational time window is needed. This means a higher temporal resolution in satellite data or the integration of complementary observational methods (e.g. Miles et al., 2017)."

**Fig1: Meltwater extent appears highly correlated with Denman's shear margins. Has any analysis been conducted in this regard?**

We have not done specific analysis for this. On the Shackleton ice shelf it is not uncommon that the whole domain experiences melt at some point in season (see also De Roda Husman et al., 2024 and Saunderson et al., 2022). The meltwater lakes accumulate in areas depending on surface topography, firn air content and the amount of melt, and it is not strange to see this in the shear margins, where the ice undergoes much deformation and there's lots of available surface depressions for water accumulation. Our observations are consistent with the observed lakes by Arthur et al. (2020) on the eastern half of the Denman glacier, included below for clarity.

- de Roda Husman, S., Lhermitte, S., Bolibar, J., Izeboud, M., Hu, Z., Shukla, S., van der Meer, M., Long, D., and Wouters, B.: A high-resolution record of surface melt on Antarctic ice shelves using multi-source remote sensing data and deep learning, Remote Sensing of Environment, 301, 113 950, https://doi.org/https://doi.org/10.1016/j.rse.2023.113950, 2024.
- Saunderson, D., Mackintosh, A., McCormack, F., Jones, R. S., and Picard, G.: Surface melt on the Shackleton Ice Shelf, East Antarctica (2003–2021), The Cryosphere, 16, 4553–4569, https://doi.org/10.5194/tc-16-4553-2022, 2022

- Arthur, J. F., Stokes, C. R., Jamieson, S. S. R., Carr, J. R., and Leeson, A. A.: Distribution and Seasonal Evolution of Supraglacial Lakes on Shackleton Ice Shelf, East Antarctica, The Cryosphere, 14, 4103–4120, https://doi.org/10.5194/tc-14-4103-2020, 2020

Figure from Arthur et al. (2020) of detected surface meltwater lakes on a subset area of the Shackleton Ice Shelf.

Fig2: What do you mean by pixel quality? Be more specific. This figure also needs a close-up on an actual drainage event, showing the related fracturing observed in the raw data (see earlier comment).

We have rephrased the caption of Figure 2, the part now reads "The colour of each dot represents the respective value at the drainage location for the respective melt season in which it occurred"

And indeed, we have provided close ups of the drainage events, but included them in Figure 1 in the manuscript (See Figure R2 in this document).

**Fig3: Include damage evolution with the tidal cycle (see earlier comment).**

We agree that a crevasse opening- and closing evolution in tandem with the tidal movement would be nice, but, as also stated in the earlier comment, it is not straightforward to include the desired damage evolution from NeRD in this way: we don't resolve individual features, and cannot for certain ascribe damage signal strength to crevasse depth, since it is derived from its visual, spatial appearances. NeRD is well suited for investigating damage patterns, but these are not extremely variable from year to year (refer to shown damage maps in Figure R1 above). We think it makes more sense to approach these temporal evolutions from a modelling based study in future research.

**Reviewer 2**

We thank the reviewer for their time and feedback on the manuscript. Their valuable comments and questions have been carefully considered, and we discuss below how we have incorporated their suggestions in the new version of the manuscript. The manuscript has undergone major revisions, which we will introduce briefly here, before providing more detailed response to the reviewer and their concerns:

- We have extended our temporal analysis to increase our sample size of observed lake drainages, from 2015-2024, adding 6 more years. We now observe 21 lake drainage events (instead of 13).
   There have been no major changes to the results by including these new events, however, we have made major textual edits to provide more nuance to the previously too strongly made claims.
- We have incorporated a comparison to the vulnerability map of Lai et al. (2020), providing insights into where they estimated the ice shelf to be vulnerable to hydrofracturing and where we observe lake drainage events. We observe quite a number of drainage events in 'non-vulnerable' areas, that are characterized mostly with relatively moderate activeness and high damage, compared to the vulnerable areas having drainage events of high activeness and high damage.
- Manuscript Figure 1 now includes examples of detected drainage events.
- Manuscript Figure 2 now includes a panel with the comparison to Lai et al.'s detected vulnerability as well as plots of data distribution that were previously part of supplementary material.
- Manuscript Figure 3 is only adjusted to incorporate the new observations.
- We have also made textual changes to clarify the methods.

**General Points**

One issue is that the study is quite limited in scope, both in terms of space and time. The generalizability of the conclusions is very limited by the fact that this study only covers three years and one relatively small ice shelf. Given the computing platform the authors used (google earth engine), it seems like it could have been relatively simple to extend the work to other locations and, within the restrictions of the datasets used, to more melt seasons.

The main novelty of the paper is the consideration of 'activeness' (i.e. how perpendicular is ice flow to the predominant fracture orientation). And one could imagine a paper on this topic (1) proposing activeness as an important factor and (1) thoroughly test if this is the case. However, without an extension in the temporal and spatial coverage of the analysis, I think it is difficult to conclude much about the importance or otherwise of activeness for lake drainage. For example, as I note below, the authors state "...it is clear that at least one of these factors, damage or activeness, must be present for lake drainage to occur." (albeit with a caveat that the exact relationship needs further investigation). I think, given the results presented here, this is a much too strong conclusion to draw at this stage. That leaves the main contribution of the paper as (1) proposing activeness as an important factor. It is worth making it clearer that this is the main contribution of the paper. Or alternatively extending the analysis so that it can achieve (2) as well. This would involve extending to other ice shelves and/or other time periods.

We recognize the limitations of our study, and understand the concerns of the reviewer as a result of that. As the reviewer describes clearly below, the main aim of this study was to assess when/where the combination of melt lakes and damage would (not) lead to hydrofracturing, and we propose activeness and tidal-induced flexure as an important factor. We concur that some conclusions might be worded too strong and have rephrased the manuscript to focus on that contribution and bring more nuance to the conclusions; more specifics are explained below. Our intent was, and remains, to present a case study to address the factors contributing to hydrofracturing/lake drainages, which is also why we chose a Brief Communication format. We think that the Shackleton ice shelf presents an ideal case study with multiple 'types' of lakes and lake drainages, as well as various damage features. We agree that a larger scale application is valuable and have extended our analysis to the time period 2015-2024 (adding 6 more years, now detecting 21 drainage events). We have not processed more ice shelves, as we think adding more ice shelves will not necessarily provide more insights, and it will take a significant amount of more processing to get to a significantly large set of observations – while doing so is very likely to overcomplicate the interpretation of the results. We therefore chose to keep this study as a contained case study of a single ice shelf.

Other important potential extensions that are missing include between lake drainage locations and ice-shelf stresses and/or vulnerability to hydrofracture as proposed by Lai et al. (2020). These are obliquely discussed

in the context of introducing the idea that how close fractures are to perpendicular to ice flow (activeness) is an important factor. However, no comparisons are made to them directly. This would seem like a more direct way (or at the least a complementary way) of testing the ideas that lie behind the activeness proposal.

This is a valuable suggestion, and have incorporated a comparison to the vulnerable regions as detected by Lai et al. (2020) in the new version of our manuscript, see below in Figure R1. The figure indicates the (non-)vulnerability to hydrofracturing of the ice shelf areas, i.e. if their observed fractures are unstable if inundated with water. Our detected lake drainage events are superimposed on the map, with the color indicating the assessment of Lai et al. (2020) at their location, showing that many events occur in an area that they assessed as resistive to hydrofracturing. We have expanded and revised our results and discussion to incorporate the interpretation of this figure.

Figure R1 Comparison of detected drainage events to the vulnerability estimates of Lai et al. (2020). This has been included as a panal in Figure 2 in the manuscript

**Specific comments**

L14: "only some of these lakes drain." This needs citations

We have added two references to support that statement:

Stokes, C. R.; Sanderson, J. E.; Miles, B. W. J.; Jamieson, S. S. R.; Leeson, A. A. Widespread Distribution of Supraglacial Lakes around the Margin of the East Antarctic Ice Sheet. *Sci Rep* 2019, *9* (1), 13823. https://doi.org/10.1038/s41598-019-50343-5.

Arthur, J. F.; Stokes, C. R.; Jamieson, S. S. R.; Rachel Carr, J.; Leeson, A. A.; Verjans, V. Large Interannual Variability in Supraglacial Lakes around East Antarctica. *Nat Commun* 2022, *13* (1), 1711. <a href="https://doi.org/10.1038/s41467-022-29385-3">https://doi.org/10.1038/s41467-022-29385-3</a>.

L17: missing "an"

Agreed and implemented.

L44: reference should not be in parentheses

Agreed and implemented.

L44: explain what "Median image mosaics" are in more detail

Agreed this could have been more clear. The sentence (**L55**) now reads "Median image mosaics are produced over time periods of 8 days for L8 and 10 days for S2 by taking the pixel-wise median value of all images within the respective period – stitching and combining individual overpasses into one domain-covering image – with resolutions of 30 m and 10 m, respectively."

L46: comma before 'which'

Agreed and implemented.

Section 2.1: Inconsistent tense. Are and was are both used. Change to be consistent.

Checked and Corrected.

L48: not clear what grid is being referred to here. Is this a spatial reference grid? If so, give its name. Or maybe it is not needed here, unless it's important for the analysis that it was this grid for some reason.

The grid mentioned here is a simple processing approach to tile the data in smaller files to streamline the processing of mosaics. It is not strictly necessary and has been removed for clarity.

L52: rephrase "Antarctic ice flow velocity observations of 2019"

We have changed the wording to (L65):

Annual ice flow velocity observations are sourced from the ITS LIVE campaign (Gardner et al., 2020).

L53: delete "and kept constant during the studied period."

Agreed and implemented.

Figure 1 caption: add 'the' before MEASURES.

Agreed and implemented.

L62: rephrase "combining all lake masks of a season."

We hope with the following we can clarify the lake extent creation.

**L68:** The location and depth of supraglacial lakes are determined using a threshold-based method on the collected mosaics.

**L76:** For each melt season, the maximum lake extent is derived by aggregating all lake masks per mosaic generated during that period.

L78: "damage maps of 300 m" should be "300m-resolution damage maps"

Agreed and implemented.

L83: I am unclear what "(tolerating a deviation of 15°)" means.

We have precised our answer and revised the text for clarity:

**L101:** Then, an area was considered likely to be 'active' if the difference between the damage orientation and the flow angle falls within  $45^{\circ} \pm 15^{\circ}$ , which occurs mainly in the shear zones of the ice shelf or in regions exhibiting mixed-mode opening fractures."

L98: delete "under-"

Agreed and implemented.

Figure 2 caption: rephrase "show NeRD damage and activeness of summer 2019/20" and "pixel quality" (I am not sure what quality refers to here).

Agreed this was unclear. The caption of Figure 2 now reads:

"Spatial occurrence of observed lake drainages on Shackleton ice shelf. a) presents the satellite-derived damage and b) the activeness metrics, both for the 2019/20 melt season. c) shows the comparison to vulnerability to hydrofracturing estimates of \citet{Lai.VulnerabilityAntarcticalce.2020}. Panels d-g show zoomed section around the glaciers on the ice shelf, with d) maximum lake extent and e-g the same as a-c. The superimposed coloured dots display the location of observed drainage events, the colour of each dot represents the respective value at the drainage location for the respective melt season in which it occurred. Thick/thin black line: Grounding and Shelf coastline from MEaSURES data"

L103: explain what "record summer" means explicitly. I am guessing it means record high temperatures and/or melt volume/extent. But which of these, I am not sure.

You are right, the term "record summer" refers to the 2019-2020 season being notable for its unprecedented melt extent and duration, record surface meltwater ponding, and anomalously high air temperatures (Banwell et al., 2021). This has been edited in the manuscript.

L104: delete 'such as those by' and put the citations in parentheses Agreed and implemented.

L105: replace 'the' with 'our'

Agreed and implemented.

L100: should this reference figure 3 as well as or instead of figure 2?

In the sentence in L100, "However, our findings indicate minimal lake formations in this region, suggesting immediate drainage of meltwater into the ocean in highly damaged areas" we mean the drainage/runoff of meltwater without first forming a meltwater lake. Therefore this is not included in Figure 3 (lake drainage events). We have added "..., without meltwater lake formation" to the sentence (L127).

L105-107: "However, the findings indicate minimal lake formations in this region, suggesting immediate drainage of meltwater into the ocean in highly damaged areas (Figure 1 and 2)." This is not clear. Which part are you referring to as the north of the ice shelf? The north-west-most part is not covered by the damage map and the region to the south of that is not ubiquitously highly damaged, so I am not sure this is a fair conclusion to reach from these two maps (damage and meltwater).

Our apologies, this was indeed unclear. We have specified we meant the area North of Masson Island, and have added the northern tip of West Shackleton to complete the maps.

L116: how did you decide on these values for the categories no, medium and high?

We see this was not clear. Different than the melt lake thresholds, there is limited previous research using similar damage detection approaches. As the damage signal is a measure of feature contrast in the satellite image, there is unfortunately not a quantitative translation to physical properties (such as crevasse depth or

density). We have therefore discretized the damage signal values to obtain a data-based estimate of what our 'low', 'medium' and 'high' values were. Due to the strongly skewed data distribution, we discretized the damage signal values in bins of unequal width, containing progressively less data samples (damaged pixels) to favor the representation of high values.

A similar approach was done to discretize the Activeness parameter. As this parameter has a normal distributed data, we divided the buckets to favor both the low and high ends of the curve.

We have clarified in the manuscript,

- L138-140: "Due to the strongly skewed data distribution we discretized the damage signal values in bins of unequal width, containing progressively less data samples (damaged pixels) to favor the representation of the minority, high values, class."
- L151: Similar as the damage values, we categorized the activeness in the following groups to favor the tails of the distribution

The distributions and threshold are visualised in Figure R2 in this document, and included in Figure 2 in the manuscript.

Figure B2: I think this would be useful to have in the main paper.

We agree and have added the distribution figures in Figure 2

Figure R2 Distribution of detected damage and activeness parameter, now included in Figure 2 in manuscript

L118: replace "a detailed listing" with "a list"

Agreed and implemented.

L127: this is not an indication of where the ice is deforming, only of where it is flowing perpendicular to fractures. The ice is actively deforming essentially everywhere.

This is correct, deformation occurs everywhere on the ice shelf. This activeness metric specifically targets areas where the relative orientation between flow and fractures, indicating dynamic activity with respect to the fracture only. We rephrased:

**L147:** The activeness parameter provides insights into the dynamic behavior of the ice shelf, with high values indicating areas where the local fractures have a high likelihood to be under active development due to the flow of the ice.

L128: replace "distributed in a bell-shaped curve" with "normally distributed" Agreed and implemented.

L131: clarify what "the glaciers," refers to here.

We referred here to the main glaciers of Shackleton Ice Shelf. We rephrased:

**L153:** "Areas close to the Northcliffe, Denman, Scott and Apfel glaciers serve as the clearest examples of areas exhibiting high activeness on Shackleton ice shelf"

L132: delete 'this'

Agreed and implemented.

L133: "tends to concentrate inland, away from the ice shelf edge" I do not see this spatial distribution in the figure. Can you explain this in more detail? What exactly do you mean by concentrate?

L136: would "fast flowing areas' be more precise than "glacier zones"? Agreed and implemented.

L138-139: "Unlike damage values, the activeness parameter does not appear to be directly related to the distribution of accumulated meltwater" This implies that the distribution of meltwater is directly related to damage. This is mentioned briefly at the start of section 3, but is this what is being referred to here? This should be made a little clearer and perhaps this statement softened somewhat, given that the distributions of damage and meltwater accumulation have not been explored in detail and it has not been established that there is a close connection (see my comment on L105-107).

The reviewer is right, this was not worded clearly, and has not been established. We have removed the sentence.

L145: In what sense is the ice shelf a prototypical example? This paper provides no comparison to other ice shelves, so if it is typical, this needs to be discussed. And I think prototypical refers to this example in some way being the originator, or the original version of something, which, unless I am missing something, it is not.

This was not meant to discuss the whole ice shelf: L145 referred to drainages A-E as 'example' where both high damage and high activeness are detected. This might be an effect to English being our second language. The sentence has been revised to (L158) "Drainages A to E on the grounding line on West Shackleton are cases where both damage and activeness are high at the lake drainage sites"

L147: I am not sure what "the glacier tongue" is referring to. Please clarify.

The Denman Glacier. Implemented

L157-159: "Although the exact relationship between these metrics requires further investigation across different ice shelves and with more drainage events, it is clear that at least one of these factors, damage or activeness, must be present for lake drainage to occur." As mentioned in the main point above, this statement should be softened. There are examples in Greenland of fracture perpendicular to the background flow direction directly draining lakes on grounded ice. It seems likely the same is possible on ice shelves. A more precise statement restricted to what this dataset tells us about this ice shelf over these three years, given the limitations of the remote sensing datasets and your analysis, is needed here. In other words, not only does the quantification of the exact relationship require further investigation, so does establishing that activeness and damage are a requirement at all.

Yes, we concur this could be more nuanced, and have done quite some revisions to achieve this.

We have revised the specific sentence:

- **L172**: "Although the exact relationship between these metrics requires further investigation across different ice shelves and with more drainage events, it seems that the activeness of detected damage adds insights into the where lake draianage occur in otherwise unsuspected areas."

**And in the discussion:**

- **L209-210**: "Nevertheless, this method complements the use of detected damage alone, and provides new insights with respect to the vulnerability to hydrofracturing previously determined based on linear fracture mechanics by Lai et al. (2020)."
- **L224-225**: "Additionally, process-based modeling studies can be used to study and resolve the physical relationship between the ice dynamics, fracture mechanics, and meltwater accumulation to the observed lake drainage."

L229-231: "This complexity underscores the need for a holistic approach when studying ice shelf
hydrofracturing. While our study provides new insights, it is important to recognize that more
process-based studies are needed to understand the full system that leads to hydrofracturing."

L161-163: I suggest deleting this opening paragraph.

Agreed and implemented.

L166-167: I suggest deleting the opening sentence of this paragraph.

Agreed and implemented.

L171: Delete "Our data reveals a trend:" A trend implies something changing over time. Also, it is unnecessary. I suggest just describing the relationship.

Agreed and implemented.

L177: delete "not only confirms but also extends the work of" and put the citation in parentheses. It's not clear to me that this analysis confirms that work. Maybe you can say that it is broadly consistent with it.

We have revised the text to clarify and simplify our findings as follows:

**L86:** "Our observations of several drainage events indicate that tidal forcing plays an important role in modulating supraglacial lake drainage across the ice shelf. These results are consistent with previous findings (Trusel et al., 2022)."

L180: delete "highlight the multifaceted nature of drainage events and" Agreed and implemented.

L182: delete "intricate"

Agreed and implemented.

L192-193: "the temporal resolution of satellite passes may not be sufficient to capture all drainage events, especially those of short duration." This is a little repetitive of earlier in the paragraph.

We have removed this redundant phrase from the paragraph.

L108-109: "produce different results" can you be more precise with this statement. Changing the thresholds would of course change the results quantitatively, but could it change things qualitatively too?

The discretization of detected damage and its activeness could change qualitatively if the thresholds are changed, depending on the data distribution. We have, however, no lack of observed values for these parameters (as opposed to observed lake drainages), and do not expect extreme changes when incorporating other regions to the data.

We have revised the text to read, **L213-216**: "The methods in this study are, in essence, transferable to other regions. However, we have tailored post-processing steps to this study area, and hence values presented in this study are specific to this ice shelf. Firstly, we normalized the detected damage and activeness values based on their observed maxima in the spatiotemporal domain. Similarly, we discretized the values into low, moderate and high classes with bin edges based on the overall data distribution, to give weight to the minority classes in our analyses. Including data from other regions might change the normalisation and discretizations of the data distribution."

L226: this sentence mentions a predictive model for the first time. It isn't clear what this is referring to. What would be the purpose of such a model? I was assuming it would be some kind of parameterization in an ice-sheet model, but the need for real-time data confusing me in that case.

We acknowledge that the introduction of a predictive model in this context may have been unclear. Our intention was to suggest that combining damage severity, activeness, and meltwater accumulation could serve as indicators for potential drainage sites. In an ideal scenario, this could be expanded to a model that predicts hydrofracturing before it occurs (hence the near real-time data). Nevertheless, we agree this is probably a bridge too far and too speculative. We have revised the sentence

L221-223: Future research could focus on developing (probabilistic) models that can identify subtle changes in ice shelf characteristics that precede drainage events, and conducting high-resolution time series observational analyses to better understand the temporal dynamics of meltwater accumulation and drainage.

The discussion: One limitation to the idea of activeness being a control is that fractures could advect into areas that are compressive, but the fractures could remain still perpendicular to flow. This scenario would yield non-zero activeness, but may not be conducive to hydrofracture. This underlines the utility of comparing lake drainage locations to ice shelf stresses and fracture orientation to principal stresses orientations.

The reviewer makes a good point, it is indeed possible that we detect false positives or false negatives. We have reduced this impact by downsampling the detected damage and its activeness, but we acknowledge that the analyses in this study are quite generalised. We fully agree that more detailed investigation of specific events are necessary to fully understand what is driving rapid lake drainages, but think that the methods in this study are not the best suited to achieve this. We have specified this more clearly in the discussion,

- L208-210. "Similarly, the activeness parameter offers a simplified approach to identify areas of active damage development, by comparing fracture orientations to the local velocity field, but is limited in providing detailed physical understanding of individual events – which would need a more thorough understanding of local ice stresses and strain rates."
- **L224-225:** "Additionally, process-based modeling studies can be used to study and resolve the physical relationship between the ice dynamics, fracture mechanics, and meltwater accumulation to the observed lake drainage."

Data availability: It would have been good to have access to the code and data for the review process.

Apologies. Google Earth Engine and Python code are now available through the links below. Please beware those are prelimary versions. For the final submission the repositories will be revisted, cleaned and prepared with documentation.

GEE: https://code.earthengine.google.com/?accept\_repo=users/juliussommer/HydrofractureShackleton

Python Github: <a href="https://github.com/js-chemE/HydrofractureShackleton">https://github.com/js-chemE/HydrofractureShackleton</a> 2023

**Reviewer 3**

We thank the reviewer for their time and feedback on the manuscript. Their valuable comments and questions have been carefully considered, and we discuss below how we have incorporated their suggestions in the new version of the manuscript. The manuscript has undergone major revisions, which we will introduce briefly here, before providing more detailed response to the reviewer and their concerns:

- We have extended our temporal analysis to increase our sample size of observed lake drainages, from 2015-2024, adding 6 more years. We now observe 21 lake drainage events (instead of 13).
   There have been no major changes to the results by including these new events, however, we have made major textual edits to provide more nuance to the previously too strongly made claims.
- We have incorporated a comparison to the vulnerability map of Lai et al. (2020), providing insights into where they estimated the ice shelf to be vulnerable to hydrofracturing and where we observe lake drainage events. We observe quite a number of drainage events in 'non-vulnerable' areas, that are characterized mostly with relatively moderate activeness and high damage, compared to the vulnerable areas having drainage events of high activeness and high damage.
- Manuscript Figure 1 now includes examples of detected drainage events.
- Manuscript Figure 2 now includes a panel with the comparison to Lai et al.'s detected vulnerability as well as plots of data distribution that were previously part of supplementary material.
- Manuscript Figure 3 is only adjusted to incorporate the new observations.
- We have also made textual changes to clarify the methods.

**General**

The goal of the work is very straightforward and well defined. Basically, there are three main goals: 1) map the overlap of damage and ponds; 2) find the draining ponds and identify in which conditions they occur; and 3) identify triggering events.

It is an interesting study, very important in the study of Antarctica. However, it is hard to define how impactful the findings are, since the methodology lacks more description of the metrics. Furthermore, the analysis should be deeper (see Major Comments).

I think the manuscript has 4 main issues:

- 1. observations do not support the claims in the manuscript
- 2. lack of a more robust statistical analysis
- 3. small number of observations
- 4. more descriptive methodology and how are the thresholds defined

We understand and acknowledge these concerns. Generally, concern 1-3 stem from the small number of observations and how these were used and analysed. For the revised manuscript we have (a) extended the time series to increase the number of observations to support our analysis and claims, to 2015-2024. The total number of drainage event is now 21 (instead of 13). The new results are incorporated in the manuscript and do not change the results significantly.

Although almost doubling the observations, it remains a small sample set for robust statistical analysis; we agree with the reviewer on this point. We have therefore (b) made efforts to rephrase our conclusions to align the observations to the claims. We think, even with the small sample size, this study can add to the scientific understanding and discussion of when/where hydrofracturing is observed, adding more nuance to a general state of vulnerability of all fractures to hydrofracturing. We provide more details in the comments of the reviewer below.

Considering point (4): we put more effort in describing and clarifying the methods. The NeRD algorithm and the thresholds specifically, both for lake detection and damage detection, are given more attention. We have specified this in the relevant comments of the reviewer below.

Google Earth Engine and Python code are available through the links below. Please beware these are prelimary versions. For the final submission the repositories will be revisited, cleaned and prepared with documentation.

GEE: <a href="https://code.earthengine.google.com/?accept">https://code.earthengine.google.com/?accept</a> repo=users/juliussommer/HydrofractureShackleton Python Github: <a href="https://github.com/js-chemE/HydrofractureShackleton">https://github.com/js-chemE/HydrofractureShackleton</a> 2023

**Major Comments**

**2.1 Sample size**

The authors used a time span of 3 years for the analysis performed. They found 13 events, which is a small sample size. Why do you not cover a larger time span to have more data? In addition, covering other regions would be beneficial. Furthermore, the definition of threshold values seems quite arbitrary, indicating that they fit only to these present specific conditions. Also, the authors highlight the need for more sophisticated statistical approaches. This is a major concern for me, because not only the dataset is small, but it also lacks meaningful statistical analysis.

We agree with the reviever that the sample size is small, and have extended the timeseries with 6 years, now spanning 2015-2024 (all available Sentinel-1 series). This has increased the observations to 21 detected drainage events, instead of 13. We don't think covering other regions is necessary to convey the main findings of this manuscript: Shackleton ice shelf is an ideal case study that features both diverse meltwater lake distribution and diverse (un)damaged ice shelf areas, and has both draining and non-draining lakes. Adding more ice shelves will not necessarily provide more insights, and it will take a significant amount of more processing to get to a sufficiently large set of observations — while doing so is very likely to overcomplicate the interpretation of the results. We therefore chose to keep this study as a contained case study of a single ice shelf.

We have included two new figures to the manuscript to help alleviate concerns on the statistical analyses: a hexbin plot, and a comprehensive comparison to the vulnerability assessment of Lai et al. (2020) – see the included figures below. These show clearly that our detected rapid lake drainages occur in areas previously thought to be invulnerable to hydrofracturing. We furthermore have made major textual revisions to nuance our claims and interpretation of the results.

Considering the thresholds: there are multiple thresholds used throughout this study.

- The lake drainage threshold (80% of volume): this threshold has been chosen consistent with previous literature, following Doyle et al.(2013), Fitzpatrick et al. (2014), Miles et al. (2017), Williamson et al. (2017), and is thus not specific to this study. These references have been specified in the manuscript (L78).
- The threshold for noise removal in the NeRD method: this has been taken from Izeboud and Lhermitte (2023) and is not specific for our study.
- The thresholds used to discretize the damage and activeness values. These indeed are more arbitrary: we have set the bin edges based on the data distribution, so that we could better compare the majority versus the minority classes. We have clarified in the text:
  - L138-140: "Due to the strongly skewed data distribution we discretized the damage signal values in bins of unequal width, containing progressively less data samples (damaged pixels) to favor the representation of the minority, high values, class."
  - L220-225: "The methods in this study are, in essence, transferable to other regions. However, we have tailored post-processing steps to this study area, and hence values presented in this study are specific to this ice shelf. [...]"

Figure R1 Hexbin plot of 'damage' and 'activeness' parameters, including the values for the detected drainage events. This is included as a panel in Figure 2 in the manuscript

Figure R2 Comparison of detected drainage events to the vulnerability estimates of Lai et al. (2020). This has been included as a panal in Figure 2 in the manuscript

**2.2 Not supportive data**

I think the data presented do not support the conclusions drawn. In fact, the authors repeatedly make an affirmation and draw it back after a few sentences, for example: L139-140 = "However, all of the detected lake drainage events occur in areas of the ice shelf classified as medium to highly active"

And L151-L153 = "For example, drainages H, K, and M took place in areas with relatively low damage but high activeness, while drainages A, F, and G occurred in the least active regions, yet

showed high levels of damage." Furthermore, analysis of Figure B1 reveals that 1 of the events is in a low activeness area, undermining the claim in L139-140.

We concede that the wording of these claims were too strong, and we apologize for any inconsistencies. With the extended timeseries and new drainage events, the discussion of these results has undergone major revisions. This section of the results reads now

L161: "Considering both damage and activeness, several notable patterns emerge. First, drainages A to F, on the grounding line on West Shackleton are cases where damage is high and activeness is moderate at the lake drainage sites. This indicates that areas along the grounding line are prone to such events due to the combined effect of significant damage and moderate activeness.

Second, east of the glacier tongue (drainages J and M), lake drainage events occurred in areas of high activeness, despite medium levels of detected damage. This suggests that the interaction between the small glaciers and the Bunger Hills creates a highly active section of the ice shelf, where accumulated meltwater rapidly drains even in moderately fractured regions. Third, we also detect a few drainages in areas with high activeness (drainages G and M) but with high levels of detected damage.

Taken together, the results suggest that while moderate damage appears to be a necessary precondition, activeness acts as an additional driver that can amplify the likelihood of drainage events. Drainages are more likely to occur if either detected damage or activeness is high, or both.".

**2.2.1 Activeness parameter**

The authors conclude that "all of the detected lake drainage events occur in areas of the ice shelf classified as medium to highly active", but they sum up to 90% of the studied region. Taking into account that they have only 13 drainage events, their conclusion is not supported by the data since further analysis should be made.

Even if drainage events are correlated to activeness, it could be the case that activeness is actually related to damage (meaning that damage is higher when it is caused by the flow, which is very plausible) and that drainage events are mainly driven by damage (also very plausible).

We completely understand the critical standpoint against the low amount of samples. With the new drainage events included in the analyses, this specific sentence has been heavily revised. In short, the results show that drainage events occur in areas with either high damage or high activeness, or both, and the text has been revised to discuss these results (For full text, refer to the result section).

- L**167**: "These results suggest that drainages can occur if either detected damage or activeness is high, or both".
- L172-173: "These drainages are in areas of both moderate and high activeness, mostly in combination with high damage areas. This highlights the role of ice dynamics in the behavior of hydrofracturing, and suggest that vulnerability estimates based only on fracture mechanics is not sufficient"

A note about the damage signal strength: this is uncorrelated to the ice flow, as it is purely based on how clear the damage feature is visible in the image, as a stark white line in a dark field on radar images (in optical images it is a dark line on white), which is mainly dependent on the look angle of the Sentinel-1 sensor and the amount of noise/speckle, where backscatter of the radar is generally highest for steep vertical walls. The strongest damage signals are actually found near and at the ice front, where large full ice-penetrating rifts yield the highest contrast between the ice and the ocean.

Apart from this, the activeness parameter is indeed correlated to damage. It is based on the orientation of the detected damage, not on the strength of the detected signal, but still, there has to be a detected damage feature to get an activeness assessment. The intent of the activeness parameter is to shift the focus of the detected damage signal strength: you can find small yet opening crevasses ('active') as well as large but stationary rifts ('passive').

This is clarified in the text, L25-28: "High activeness, indicative of crevasse opening, would facilitating new routes for lakes to (suddenly and rapidly) drain, whereas more passive crevasses either prevent the formation of lakes by providing direct runoff for meltwater or might remain stable when inundated with meltwater."

**2.2.2 Section "Lake Drainage Events in Periods of Increasing Tidal Heights"**

As reviewer #1 said "6/11 drainage events in 2019 (more than half) started during the lowest (or even descending) phase of the tidal cycle (drainages M, L, F, H, J, E)". This simply invalidates the sentence: "Our findings unveil a compelling narrative of ice shelf dynamics, revealing an intricate interplay between tidal forces and supraglacial lake drainage events".

Furthermore, in a hypothetical case where all the drainage events occur in an ascending amplitude phase of the tidal, it will not imply what is said: if you have thousands of lakes in a ice shelf, and each time only a small fraction of lakes drain, the likelihood of a drainage event triggered by tidal flexure would be higher in the crest of the amplitude phase.

We refer to our response to reviewer #1, which we will repeat here as well: we would like to clarify that indeed drainage events M, E and K 'start' in the descending phase it is important to realise that the drainage events shown in figure 3 occur *between* the detected dates t1 and t2, since we only detect 'lake is present' at t1 and 'lake has drained' at t2 – it is not a draining that starts at t1 and ends at t3. Therefore, Figure 3 does (in our opinion) suggest that drainage does not seem to occur in the descending phase. This is further corroborated with the new drainage events.

We do acknowledge that we cannot differentiate if the drainage occurs in the lowest part of the cycle or the ascending part, and that we cannot pinpoint the exact moment of drainage, so have adjusted the text accordingly:

**L184-186**: "We compare the drainage time-windows to tidal data (Figure 3), and indeed find a clear pattern: the majority of drainage episodes aligns with the ascending phase of tide cycles. Although we cannot determine the exact drainage date, only a snapshot before and after the event, few drainages seem to have occurred on the descending phase"

As for the higher likelihood of a drainage event triggered in the crest of the amplitude phase: this is not really reflected in our observed drainages, with the exception of two events (O and D).

**Minor comments**

**3.1 Methodology clarification**

Some clarifications are needed on the two main metrics used in the study. First, I agree with reviewer #1 about the lack of information on how damage is calculated. Usually it is inverted from  $\mu = (1 - D)B2\varepsilon$  n-1ne (1)

where  $\mu$  is the ice viscosity, D is damage (which you want to invert), B is the ice rigidity,  $\epsilon$  is the effective strain rate, and n is the flow law exponent. I think it is needed to specify how damage is calculated from remote sensing. Also, if possible, it would be interesting to relate the damage calculated in the present work (which reviewer #1 suggests changing the name to "satellite-derived damage" and I agree), and damage calculated from Equation 1.

We see that in lieu of brevity the methods have been too concise, and will add more details on how exactly damage is calculated: L89-92: "In short, the NeRD method consists of the following steps: (i) create cut-out windows from the image, (ii) apply the Normalised Radon transform to these windows, (iii) extract dominant feature signal strength and orientation for every window, (iv) quantify the damage signal by removing noise from the signal and (v) postprocessing. In the post-processing step we clipped the product to the ice shelf bounds"

As the NeRD method is a published algorithm (Izeboud and Lhermitte, 2023), the description will remain short and to the point, as for extended sensitivity studies and evaluation of the method we refer to that publication. Nevertheless, we will include the produced annual damage maps (example included below, before downsampling to 3 km) for every year in the supplementary material.

We agree with the suggestions to clarify we use satellite-derived damage, and will implement this term as well as referring more strictly to 'damage signal' or 'detected damage' in the manuscript.

- **L21-22** "Specifically, we propose that in addition to the presence of visible damage (open crevasses, fractures, and rifts), a measure of the `activeness' of the damage feature (i.e. crevasse opening or propagation) can be used..."
- L42 "These drainage events are compared to satellite-derived damage and..."

Lastly, we agree that it would definitely be interesting to relate the calculated damage maps to damage calculated from damage mechanics models, and to our knowledge this is an active field of research – refer to e.g. Gerli et al. (2024) and De Rydt et al. (2021) – but we consider it out of scope for this study.

- Izeboud, M. and Lhermitte, S.: Damage Detection on Antarctic Ice Shelves Using the Normalised Radon Transform, Remote Sensing of Environment, 284, 113 359, https://doi.org/10.1016/j.rse.2022.113359, 2023.
- Gerli, C., S. Rosier, G. H. Gudmundsson, and S. Sun. 2024. 'Weak Relationship between Remotely Detected Crevasses and Inferred Ice Rheological Parameters on Antarctic Ice Shelves'. The Cryosphere 18 (6): 2677–89. https://doi.org/10.5194/tc-18-2677-2024
- De Rydt, J., R. Reese, F.S. Paolo, and G. H. Gudmundsson. 2021. 'Drivers of Pine Island Glacier Speedup between 1996 and 2016'. Cryosphere 15 (1): 113–32. https://doi.org/10.5194/tc-15-113-2021

Figure R3 Example of annual damage maps, detected by the NeRD method at 300 m resolution, before downsampling to 3 km

**3.2 Activeness vs. damage**

I think a further analysis of the relationship between damage and activeness is required. It can be the case that the relationship between them is high, so any relationship between activeness and drainage occurrences is only due to damage.

The activeness parameter is correlated to damage, but only in the sense that it is determined in areas where damage is detected (binary): it is based on the orientation of the detected damage, not on the strength of the detected signal. It shifts focus to damage features that are likely undergoing change (opening, widening) from features that are stationary/passive.

Furthermore, apart from finding drainages where activeness is high, we also find the opposite: damagedareas with low activeness do not facilitate any of the observed drainage events. It is an interesting point though, and there could be areas of 'activeness' without (yet) visible damage features to show for it. However, since the damage maps are quite consistent in the detected patterns from year to year, we presume this to be unlikely.

**3.3 Motivation**

The first sentence of the article ("Surface lake drainage can destabilize ice shelves, occurring either slowly via supraglacial channels or rapidly through crevasses") makes me wonder if you are not studying the opposite. Instead of analyzing how damage influences lake drainage, should not you analyze how lake drainage influences damage? Otherwise, if you really want to analyze how damage influences lake drainage, you should motivate that in the introduction. I think you are putting the cart before the horse.

Thank you for this insight, it's important to us that the introduction is very clear. We are mainly analysing the place and timing of lake drainages, and in that sense we are analysing how damage influences lake drainages: we hypothesised that just 'having' damage features does not necessarily lead to hydrofracturing – since damage is so abundent on many ice shelves in antarctica, and hydrofracturing is less widespread.

We have clarified in the introduction, e.g. **L15-18**: "However, given the widespread presence of crevasses and other damage features on Antarctic ice shelves, it remains unclear to what extent pre-existing damage influences the likelihood and timing of lake drainage events. Here, we use observations of lake drainage events from remote sensing data to study their place and timing, examining whether damage alone is sufficient to indicate a potential of hydrofracturing, or if additional conditions, such as tidal forcing, are necessary to initiate lake drainage."

**Specific comments**

L16: Is it the first time that "activeness" is used? If so, say that the manuscript introduce this concept. Otherwise, make a reference.

The concept of activeness is newly introduced by us. And has been clarified in the text:

L15: We therefore hypothesize that, apart from using the presence of damage features, another metric is needed to indicate a likelihood for occuring lake drainages. Specifically, we propose that in addition to the presence of damage (open crevasses, fractures, and rifts), a measure of the activeness of the damage feature (i.e. crevasse opening or propagation) can be used to identify where lake drainages are likely to occur on an ice shelf.

**L44: Do you advect the features when you merge the images in the mosaic?**

We did not account for advection when creating the mosaics. Since each mosaic covers a maximum of 10 days, the only area where advection would be significant is at the fast flowing Denman Glacier (max speeds of ~1500 m/yr (Miles et al. 2020) in its center, ~4.6 m/day, just enough to have 1 pixel of advection in the 30 m Sentinel-2 images in the selected period) -- ice flow speed quickly drops to
- Miles, K. E., Willis, I. C., Benedek, C. L., Williamson, A. G., and Tedesco, M.: Toward monitoring surface and subsurface lakes on the Greenland Ice Sheet using Sentinel-1 SAR and Landsat 8 OLI imagery, Front. Earth Sci., 5, 1–17, https://doi.org/10.3389/feart.2017.00058, 2017
- Williamson, A. G., Arnold, N. S., Banwell, A. F., and Willis, I. C. (2017). A Fully Automated Supraglacial lake area and volume Tracking ("FAST") algorithm: development and application using MODIS imagery of West Greenland. Remote Sens. Environ. 196, 113–133. doi: 10.1016/j.rse.2017.04.032

L67: Was not the are threshold 1800? Furthermore, why do you use these threshold? Give a reason and use the citation. Only the citation is not enough.

We use two sets of thresholds in our analysis. The 1800 m² threshold is applied to minimize noise by excluding very small features that are likely spurious. In contrast, the 54,000 m² threshold is used to focus on lakes that are large enough to potentially drain and impact the ice shelf. This 54,000 m² value corresponds to about 60 pixels in Landsat 8 imagery, a size considered significant for water volume and hydrological impact (Williamson et al. (2018a)).

Williamson, A. G.; Banwell, A. F.; Willis, I. C.; Arnold, N. S. Dual-Satellite (Sentinel-2 and Landsat 8) Remote Sensing of Supraglacial Lakes in Greenland. The Cryosphere **2018**, 12 (9), 3045–3065. https://doi.org/10.5194/tc-12-3045-2018 a.

L72: you excluded 20 out of 25, so you have 5 drainage events. How then you have 13 drainage events in your results?

Apologies, but we are unsure where the count of 20 excluded events comes from. The manuscript stated "Twelve out of twenty-five events are removed, as they are judged to be refreezing lakes rather than draining lakes." This resulted in 13 events.

L83: Missing citation. Makes sense to compare the angle of the fracture to its orientation, but quantifying it can be tricky. Is there any supporting studies for the use of that values?

We apologise for any confusion: the angle of the damage feature is a result of the NeRD method (specified just below this sentence). We will clarify in this sentence ("The obtained damage orientation from the NeRD algorithm (Izeboud and Lhermitte, 2023) is used to identify areas with a likelihood of active damage development, ..."). Moreover, as we will expand the explanation of the NeRD algorithm to provide more clarity on how damage (and its orientation) is detected in the method section beforehand, this will further aid clarity.

L88: If activeness is binary, how you produce an image like Figure 2 b)? It do not seems like a product of downsampling.

Our activeness metric is initially binary (0 or 1) at the original 300 m resolution. When we downsample by a factor of 10 using an average resampling method, each 3000 m pixel then represents the average (or proportion) of active pixels in that block, which naturally yields continuous values between 0 and 1. This has been clarified in the text, L111-113: "Both 300 m maps are downsampled with a factor of 10 using an average resampling method, and normalized with their respective maxima, resulting in 3000 m resolution rasters with values between 0 and 1."

**L88: I don't see the reason of donwsampling it.**

It can be very tricky to properly assign which damage features should be linke to which lake drainage events based from these remote sensing observations, and what the appropriate lengthscale of such influence is. Furthermore, we hypothesize that an area of active damage might be indicative of a general structurally weakened ice zone, which might facilitate lake drainages through previously undetected (small) fractures. To capture this, we translated the detected damage and activeness parameter into a less localized representation, reflecting the overall integrity of the ice over a larger area.

This has been clarified in the text, **L107-110**: "From these observational products we cannot prove causality between individual damage features and specific drainage events. Furthermore, it's possible for the drainage to occurs through a fracture that is not visible in the 300 m maps. We therefore use the damage and activeness maps as an indication of a general structurally weakened ice zone, which we hypothesize to favor lake drainages through undetected or new (small) fractures. For this reason, we downsampled the damage and activeness maps to inspect the overall integrity of the ice for a larger area surrounding drainage events."

L89: Why normalizing? You can say directly that damage varies between 0 and 1 and everything is already normalized.

We are using the NeRD algorithm (Izeboud and Lhermitte, 2023) which outputs damage values between 0 and 0.5. We then normalize these values using the maximum derived from the Shackleton Ice Shelf, which standardizes the data into a 0 to 1 range for easier comparison across the study area.

L89: Add "resolution" after "3000 m".

Agreed and implemented.

L101: Add "total" in "with total maxima".

Agreed and implemented.

L116: I think the definition of the thresholds should go to methods with a further explanation of the threshold used.

Agreed, we concur that for the sake of brevity we left out too much, and we will implement this.

L119-124: This is a very sounding result, supporting the hypothesis of a strong relationship between drainage events and damage. However, I would expect the same analysis regarding the activeness. The distribution of 10%, 71%, and 19% does not allow this analysis. Contrasting the area distribution and Figure B1, and do not see a strong relationship between activeness and drainage events. I would say that most of the signal of drainage events are due to damage.

The activeness parameter was discretized differently than the damage signal due to it showing a more normalised-distribution. This has been clarified by including the distribution & hex-bin plot in Figure 2 (see Figure R1 in this document). The bins have also been made more uniform by discretizing by proportions of 20%-60%-20% of the total dataset.

**L129: Same comment as for damage.**

Clarification has been included, L151: "Similar as the damage values, we categorized the activeness in the following groups to favor the tails of the distribution: ... "

**L139: This conclusion is not surpirse. It sums 90% of the studied area.**

The reviewer is correct. We have shortened this sentence to state "indicating that areas without active damage development are not accommodating lake drainage" and have focussed the rest of the discussion on activeness on how it can provide more insights compared to fracture mechanics only (L171-179)

L151: What do you mean by "parallel trend". Be more specific. L151-153: This goes against the phrase: "However, all of the detected lake drainage events occur in areas of the ice shelf classified as medium to highly active". L156: You previously said the opposite. L158: As far as I understood, it is impossible to drainage event to occur without damage. Be more precise with this statement.

We agree that the wording was not clear. This particular paragraph has been removed and replaced by the following:

**L171-179:** Compared to the vulnerability metric of Lai et al. (2020), which indicates the (in)stability of detected fractures to inundation with meltwater, we see that drainages that occur on `vulnerable' areas also have high activeness. Intriguingly, we also detect drainages that occur in areas where Lai et al. (2020) indicated `no-hydrofracturing'. These drainages are in areas of both moderate and high activeness, mostly in combination with high damage areas. This highlights the role of ice dynamics in the behavior of hydrofracturing, and suggest that vulnerability estimates based only on fracture mechanics is not sufficient.

L161: I agree with reviewer #1 regarding the drainage events with respect to the ascending phase of tidal cycles.

We understand the concern and hope the new drainage events corroborate this more strongly. We repeat our answer from earlier in this document:

We would like to clarify that indeed drainage events M, E and K 'start' in the descending phase it is important to realise that the drainage events shown in figure 3 occur between the detected dates t1 and t2, since we only detect 'lake is present' at t1 and 'lake has drained' at t2 – it is not a draining that starts at t1 and ends at t3. Therefore, Figure 3 does (in our opinion) suggest that drainage does not seem to occur in the descending phase. This is further corroborated with the new drainage events.

We do acknowledge that we cannot differentiate if the drainage occurs in the lowest part of the cycle or the ascending part, and that we cannot pinpoint the exact moment of drainage, so have adjusted the text accordingly:

L184-186: "We compare the drainage time-windows to tidal data (Figure 3), and indeed find a clear pattern: the majority of drainage episodes aligns with the ascending phase of tide cycles. Although we cannot determine the exact drainage date, only a snapshot before and after the event, few drainages seem to have occurred on the descending phase"

L187: I think you do not have drainage events that last hours. If this is the case, remove the "few hours". You are right and we'll remove this.

L190: Here you say that it is difficult to assign the drainage events to hydrofracturing, but in the discussion you did this.

Thank you for your observation. We acknowledge that our previous wording may have been unclear. While we can identify the occurrence of drainage events (i.e. detecting a drained lake) using satellite imagery, we don't know the exact timing and speed of the drainage that occurred between the satellite overpassess. This is the distinction we were trying to convey, and we'll edit the text to clarify this.

L197: If NeRD can not identify individual fractures, how then you measure the orientation of the fractures to calculate the activeness?

Apologies, this wording is misleading. NeRD returns one value for damage signal strength and one for damage orientation for every processing window of 10x10 pixels (30 m per pixel). It does not, however, return the exact location of the detected feature within the window, and neither its width or length. It is also possible there are multiple crevasses within the window, for which case the algorithm favors the feature with the strongest contrast (see Figure from Izeboud and Lhermitte (2023) below). So, what we mean is actually that NeRD does not detect the exact outlines of individual features.

Fig. 2. Idealised scenario's to illustrate the differences between the Radon transform with and without normalisation. Panel a-e represent different scenario's to which the Radon transform is applied: a1-e1 show an idealised window with a hypothetical crevasse, a2-e2 show the corresponding 2-D feature space  $R(\rho,\theta)$  without and with normalisation (respectively top and bottom), and a3-e3 the signal response  $\sigma(\theta)$  with and without normalisation — from which  $\sigma_{crev}$  is extracted (black dot).

**214-216: This sentence is another major concern for this study.**

We understand the concern, and think this section could have been more clear. It has been revised:

**L206-211**: "We have resampled these parameters to provide an indication for larger-scale weakening of the ice surrounding the drainage events, since the NeRD method does not resolve individual fractures. This allowed us to provide a generalised comparison across the ice shelf, but limits the attribution of lake drainages to specific features and so limits a more detailed representation of individual events."

I would appreciate a hexabin graph with activeness and damage in the axes. This would allow us to see the correlation between both metrics and the occurrence of drainage events.

Excellent suggestion and this has been included, as shown in Figure R1 in this document; added to Figure 2 in the manuscript.

Table A1: Bring it to the main body of the text, it is too important. I suggest including two more columns: Classification of damage and activeness (low, medium, high).

Good idea, the extra columns have been added. We agree that it would be good to have this in the main body. However, the Brief Communications format only allows for three display items (tables/figures) so we are very limited in our flexibility here, unfortunately.

Figure 1: Define LIMA as an optical imagery mosaic from Landsat. Agreed and implemented.

Figure 2: Add ", respectively" at the end of the first sentence. Agreed and implemented.

---

## Author Response (AR2)

**Response to Review "Brief Communications: Tides and Damage as Drivers of Lake Drainages on Shackleton Ice Shelf"**

We thank the editor and reviewer for this new round of review of our manuscript. We were happy to hear that the new version of the manuscript was received well. Remaining comments or questions are discussed and implemented as detailed below (with the reviewer's comment in blue). There are no major changes to the manuscript.

Yours sincerely,
Julius Sommer, on behalf of all co-authors.

**Reviewer 1**

This manuscript investigates the role of pre-existing ice shelf damage and tidal flexure in controlling supraglacial lake drainage on the Shackleton Ice Shelf. Using NeRD-derived damage and activeness maps, the authors compare the spatial distribution of fractures with lake locations and drainage events, and further analyze drainage timing relative to tidal cycles. They find that lake drainages predominantly occur in areas classified as medium to highly damaged and active, and that drainage events often coincide with the ascending tidal phase.

The authors have engaged constructively with Reviewer 1's comments. However, several methodological choices and interpretations remain unclear or potentially misleading (particularly concerning the relative nature of the damage classification). For these reasons, the manuscript still requires further clarification and refinement before it can be published.

**General Comments:**

First, the manuscript explains that damage values were normalized to the observed maxima and discretized based on their skewed distribution, with thresholds specific to the Shackleton Ice Shelf (Section 2.4, Table B1). This is clearly acknowledged in the discussion, where the authors note that values and thresholds would need to be adjusted for other ice shelves. However, labeling these categories as "low/medium/high damage" risks suggesting that they represent physically absolute levels of structural weakening, whereas in reality they are relative, dataset-specific bins. These binnings (low/medium/high) is based on statistical groupings derived from the Shackleton distribution, NOT on any fracture mechanics threshold. Therefore, using terminology such as "high damage" risks over-interpreting these classes as representing absolute fracture intensity, which in turn, bias the physical interpretation of drainage processes. I would suggest rephrasing throughout to avoid implying a direct physical meaning — for example, by emphasizing that the classes reflect relative signal strength within Shackleton rather than absolute damage levels.

We thank the reviewer for this valuable suggestion and fully agree that the current terminology could imply an absolute physical meaning. We have refined the wording throughout the manuscript to emphasize that the "low-medium-high" categories represent relative levels within the Shackleton Ice Shelf dataset, rather than absolute measures of structural weakening.

For simplicity and visual clarity, we have retained the existing "low-medium-high" labels in the figures and tables, but have explicitly clarified in the corresponding captions that these categories reflect relative, dataset-specific groupings based on the Shackleton distribution.

L153: "We categorized damage levels into three distinct groups with values specific to Shackleton ice shelf: not/low damaged (...), medium-damaged (...), and highly damaged (...)."

And throughout text, for example: L160 and L164 "... lake drainages in **relatively** medium-damaged regions...", "... no lake drainages have been recorded in **relatively** low-damaged areas..."

A second methodological (also highlighted by another reviewer) issue concerns the lake mapping thresholds, that are still not properly explained. The classification of drainage events requires that lakes drain by at least 80%, citing a previous study. However, this threshold is never properly justified in terms of the present analysis. Requiring such a high threshold may exclude events where lakes partially drain (e.g. 50%), which could still be linked to hydrological connections and evolving damage, and would further strengthen the manuscript. The choice of 80% should be justified, and the potential implications of omitting partial drainage events should be discussed.

We understand the reviewer's point, and indeed partial drainages may also be linked to the same hydrological and damage connections as 80-100%-draining lakes. It becomes, however, much more difficult to confidently distinguish true positive from false positive detected lake drainages in the observations. Already for the 80%-draining lakes, we discard significant amount of the supposedly drainages: see Table R1 below.

Tab R1: Detected drainage events for different shrink criteria in area% in comparison to the initial lake size

|      | 50% | 70% | 80% | 80% + manual |    |
|------|-----|-----|-----|--------------|----|
| 2016 | 72  | 54  | 49  |              | 7  |
| 2018 | 22  | 14  | 12  |              | 1  |
| 2019 | 132 | 96  | 80  |              | 15 |
| 2020 | 41  | 34  | 32  |              | 2  |

For example, in the year 2019, using a 50% threshold yields 132 initially detected events versus 80 events for the 80% drainage threshold. However, after manual inspection, we found the same number of drainages (15) for both thresholds.

Therefore, the use of the 80% threshold is twofold: (a) we want to be confident in the detected event actually present lake drainage (versus refreezing or changes in snow cover) and (b) be consistent with respect to other literature (as was already discussed in the previous review round).

We have specified this in the text at L80: "While lower thresholds (e.g. 50–70 %) can also indicate partial drainage, our observations suggest that these cases are often ambiguous and more difficult to classify confidently as true drainage events, as surface changes may equally reflect refreezing, snow cover variations, or meltwater redistribution. Therefore, we retain the 80 % criterion as a conservative and literature-consistent threshold."

See below two examples of lakes that lose between 50% and 80% of their surface area, but were subsequently not classified as true positive drainage event, as inspection by our team attributed both changes to refreezing/snow cover change/runoff.

Example of 50% lake drainage, discarded as True drainage, because of suspected surface runoff.

Example of 70% lake drainage, discarded as True drainage, because of suspected surface runoff and/or refreezing.

Following other literature using 80% drainage threshold: Doyle et al. (2013), Fitzpatrick et al. (2014), Miles et al. (2017), Williamson et al. (2017).

- Doyle, S. H., Hubbard, A. L., Dow, C. F., Jones, G. A., Fitzpatrick, A., Gusmeroli, A., Kulessa, B., Lindback, K., Pettersson, R., and Box, J. E.: Ice tectonic deformation during the rapid in situ drainage of a supraglacial lake on the Greenland Ice Sheet, The Cryosphere, 7, 129–140, <a href="https://doi.org/10.5194/tc-7-129-2013">https://doi.org/10.5194/tc-7-129-2013</a>, 2013
- Fitzpatrick, A. A. W., Hubbard, A. L., Box, J. E., Quincey, D. J., van As, D., Mikkelsen, A. P. B., Doyle, S. H., Dow, C. F., Hasholt, B., and Jones, G. A.: A decade (2002–2012) of supraglacial lake volume estimates across Russell Glacier, West Greenland, The Cryosphere, 8, 107–121, https://doi.org/10.5194/tc-8-107-2014, 2014
- Miles, K. E., Willis, I. C., Benedek, C. L., Williamson, A. G., and Tedesco, M.: Toward monitoring surface and subsurface lakes on the Greenland Ice Sheet using Sentinel-1 SAR and Landsat 8 OLI imagery, Front. Earth Sci., 5, 1–17, <a href="https://doi.org/10.3389/feart.2017.00058">https://doi.org/10.3389/feart.2017.00058</a>, 2017
- Williamson, A. G., Arnold, N. S., Banwell, A. F., and Willis, I. C. (2017). A Fully Automated Supraglacial lake area and volume Tracking ("FAST") algorithm: development and application using MODIS imagery of West Greenland. Remote Sens. Environ. 196, 113–133. doi: 10.1016/j.rse.2017.04.032

Finally, there is a conceptual contradiction in the interpretation of drainage controls. The manuscript concludes that drainages occur during the ascending phase of the tidal cycle, when surface crevasses tend to close but basal crevasses are expected to open. This would suggest that basal fractures play an important role in enabling drainage. However, NeRD is known to be less sensitive to basal fractures (as evidenced by the low detected damage in the shelf interior, where basal fracturing is in fact widespread). It is therefore unclear how the presented NeRD-based damage fields can be used to support conclusions about processes that are likely governed by basal fracturing. This limitation should be acknowledged more directly, and the apparent tension between results (drainages linked to ascending tide) and method (surface-sensitive NeRD) discussed explicitly. Yes indeed, the results imply that basal crevassing plays a key role, whilst NeRD is not specifically suited to detect basal fractures. We agree with the reviewer that the damage fields as detected by NeRD cannot capture the full process that governs lake drainages. Nevertheless, we think the surface damage fields, and also the activeness parameter derived from them, do provide valuable insights on where hydrofracturing occurs – since basal fractures also don't explain the whole process (as they're widespread in the ice shelf interior, where almost no surface damage nor lake drainages are detected)

We have made this limitation more clear in the manuscript, and more explicitly discuss the importance of basal crevasses.

L246: "During the ascending tidal phase, tensile stresses at the base of the ice shelf promote the opening of basal crevasses, which facilitates water drainage when the base of the lake or a surface crevasse is reached. However, with the NeRD framework, we are not able to distinguish basal from surface crevasses to further study this process, and so cannot determine why basal crevasse opening would be more favorable to facilitate lake drainages. In line with this, future work should target and constrain the role of basal fracturing in lake formation and meltwater drainage"

L256 "To study individual drainage events in high detail, other fracture detection methods are advised, such as segmentation approaches (Surawy-Stepney et al., 2023, e.g.,) that can delineate individual features, or ground-penetrating radar observations to include basal crevasses."

**Specific Comments**

• L80 (Lake drainage definition): Why was the 80% threshold chosen? Even if a lake drains partially (or does not drain fully), such an event could still indicate hydrological connections with damage evolution. Please clarify.

See also our response to the general comments above. The use of the 80% threshold is twofold: (a) we want to be confident in the detected event actually present lake drainage (versus refreezing or changes in snow cover). Lowering the drainage threshold does not yield more clear detected drainage events (after manual inspection is performed). And (b) be consistent with respect to other literature (as was already discussed in the previous review round).

• L95 (Comparison of local damage orientation with ice flow angle): Ice flow direction can be noisy due to velocity data errors. Was the velocity field filtered or smoothed before comparison? Please specify.

We thank the reviewer to point that out. The utilized ice velocity data (spatial resolution of 240 m) was down-sampled to a resolution of 300 m, using an average resampling method. No further processing of the velocity data was performed during this study. The following figure demonstrates the smooth velocity fields used for the comparison with the local damage orientation.

We have added the figure to the appendix and further specified the down-sampling in the method section:

L64: "The ice flow velocity data is downsampled from its original resolution of 240 m to a resolution of 300 m using an average resampling method."

• L107: Add "new small fracture and basal crevasses" for completeness, since basal fracturing is also mentioned later as a possible pathway.

Agreed and implemented.

• Section 2.5 (Tidal heights): Please clarify where tidal heights are calculated. Do you compute the tidal signal at a specific offshore location, or across the model grid domain? Are there spatial variations in tidal amplitude over the study region that could affect results?

We thank the reviewer for pointing that out. Initially, we computed the tidal amplitude at the location of each individual lake. The following figure presents the amplitudes at all 15 lake locations from the Antarctic summer of 2019–2020. We came to the conclusion that despite the slight variations, it is sufficient to display only one of the tidal amplitude evolution per year in Fig 03.

These results are in the line with Padman et al. (2018), in which they demonstrate a limited variability of tidal amplitude along the Shackleton ice shelf.

We include this figure to the appendix, and further specify in L123: "Due to low tidal amplitude variations across Shackleton ice shelf (section C1, (Padman et al., 2018)), the tidal amplitudes for this study are determined at the central location of lake I (Figure 1)."

• L120: Wording issue — change "with a total surface maxima of 234 km2" to "with a total surface maximum extent of 234 km2".

Agreed and implemented. We further recalculated the total surface maximum extents by only accounting for the detected meltwater on the ice shelf (excluding the meltwater on grounded ice).

L131: "During the study period, extensive supraglacial lake extents were detected, with a total surface maximum extent of 150 km2 for 2016-2017, 213 km2 for 2018-2019, 215 km2 for 2019-2020, and 152 km2 for 2020-2021."

• L120: Consider adding a seasonal map figure (similar to Figure 1) for each melt season to help the reader visualize interannual variability.

We have added a figure of the extent of all years to the appendix:

•L135: Since NeRD does not directly detect individual crevasses, I suggest rephrasing to match your rebuttal letter: e.g., "the damage maps represent the likely presence of large fractures (>100 m) within a 300 m region." Please also add that the algorithm is more sensitive to surface crevasses, as evidenced by the low damage signal in the shelf interior where basal fracturing is expected.

We've clarified these two points in the manuscript, where the NeRD method is introduced: L98: "By using the NeRD method, we detect features at the ice shelf surface only – all visible crevasses, fractures and rifts – and group these under the umbrella term `damage', distinguishing from the damage parameter commonly used in continuum mechanics literature (Sun et al., 2017). Basal crevasses are not explicitly included, but might be detected if their concurrent surface depression is distinct. The damage maps represent the presence of large features (roughly >100 m) within the 300 m range, rather than the delineation of the feature itself."

• L140: The classification of damage into "low/medium/high" is based on the data distribution rather than physically interpretable thresholds. While I understand the need to emphasize the minority high-damage pixels, this makes the classes relative to the Shackleton Ice Shelf dataset and not absolute. To be clear, what is categorized as "high damage" here may not represent high damage elsewhere. This raises two concerns: (i) whether interpreting drainage events as occurring in "high damage" areas may be biased, since the thresholds are not physically defined, and (ii) whether results are transferable to other ice shelves without rescaling or renormalization. Could the authors clarify explicitly that these categories are relative, not absolute, and discuss how this affects the physical interpretation of lake drainage processes?

We understand these concerns, and refer to our response at the general comments. We've clarified in the manuscript that these classes are relative to Shackleton. Nevertheless, compared to damage maps in the Amundsen Sea Embayment (Izeboud and Lhermitte, 2023), the range of initial detected damage values is similar. Furthermore, we have defined that these bins are chosen to favour the 'minority class' and not to imply physical differences. The high damage signals will always be in the minority class, so even though the exact distributions might differ, we do not expect fundamental changes in results. But, naturally, they should be assessed with care when transferring this method to other ice shelves.

- L144: The statement "lake drainages predominantly occur in areas classified as medium to highly damaged" is partly a result of the statistical classification rather than a physically meaningful threshold. Please clarify this. Agreed and implemented by adding "relatively", as discussed under the earlier comment.
- L208: Similarly, the link between drainage and "high damage" is again an artifact of the classification scheme. Please emphasize this limitation.

Agreed and implemented by adding "relatively", as discussed under the earlier comment.

• L225: If lake drainages coincide with the ascending tidal phase (when surface crevasses are expected to close), this would imply basal crevasses may play a key role. Could the authors explain in more detail how basal opening could trigger surface lake drainage?

Yes indeed, it implies that basal crevassing plays a key role. During the ascending tidal phase, tensile stress at the base of the ice shelf promotes the opening of basal crevasses, which may connect to surface crevasses or the base of a lake, facilitating lake drainage. This is in essence the same process for surface fractures. Unfortunately, the reason why drainages would then occur more frequently for basal crevasse opening than surface crevasse opening cannot be inferred from our results as we do not distinguish basal crevasses explicitly, nor are we with certainty detecting purely surface crevasses. The NeRD method picks up on some basal crevasses due to the clear localised surface depression (Izeboud and Lhermitte, 2023). Therefore, we think other methods would be more suited to study this in more detail.

We have added this in the discussion, L246: "During the ascending tidal phase, tensile stresses at the base of the ice shelf promote the opening of basal crevasses, which facilitates water drainage when the base of the lake or a surface crevasse is reached. However, with the NeRD framework, we are not able to distinguish basal from surface crevasses to further study this process, and so cannot determine why basal crevasse opening would be more favorable to facilitate lake drainages. In line with this, future work should target and constrain the role of basal fracturing in lake formation and meltwater drainage"

• L230: Please expand the discussion of NeRD limitations based on your response to previous comments. Specifically, you said that NeRD provides regional likelihoods of fracture occurrence, not delineation of individual crevasses and the method is more sensitive to surface features, missing basal fractures. Would more classical segmentation approaches (e.g., Surawy-Stepney et al., 2024) improve crevasse mapping? Since basal crevassing is suggested as a trigger, future work might target methods explicitly designed to map basal features.

We appreciate the reviewer's suggestion to expand on the limitations of the NeRD method. We note thatsegmentation approaches, such as Surawy-Stepney et al. (2023), would not overcome the limitations completely. They do delineate individual fractures, but they likewise rely on surface imagery and therefore remain insensitive to basal features.

We have added the following to the discussion: L256 "To study individual drainage events in high detail, other fracture detection methods are advised, such as segmentation approaches (Surawy-Stepney et al., 2023, e.g.,) that can delineate individual features, or ground-penetrating radar observations to include basal crevasses."